# BATCH MULTIVALID CONFORMAL PREDICTION

## ABSTRACT

We develop fast distribution-free conformal prediction algorithms for obtaining *multivalid* coverage on exchangeable data in the batch setting. Multivalid coverage guarantees are stronger than marginal coverage guarantees in two ways: (1) They hold even conditional on group membership—that is, the target coverage level $1 - \alpha$ holds conditionally on membership in each of an arbitrary (potentially intersecting) group in a finite collection $\mathcal{G}$ of regions in the feature space. (2) They hold even conditional on the value of the threshold used to produce the prediction set on a given example. In fact multivalid coverage guarantees hold even when conditioning on group membership and threshold value simultaneously.

We give two algorithms: both take as input an arbitrary non-conformity score and an arbitrary collection of possibly intersecting groups $\mathcal{G}$, and then can equip arbitrary black-box predictors with prediction sets. Our first algorithm `BatchGCP` is a direct extension of quantile regression, needs to solve only a single convex minimization problem, and produces an estimator which has group-conditional guarantees for each group in $\mathcal{G}$. Our second algorithm `BatchMVP` is iterative, and gives the full guarantees of multivalid conformal prediction: prediction sets that are valid conditionally both on group membership and non-conformity threshold. We evaluate the performance of both of our algorithms in an extensive set of experiments.

## 1 INTRODUCTION

Consider an arbitrary distribution $\mathcal{D}$ over a labeled data domain $\mathcal{Z} = \mathcal{X} \times \mathcal{Y}$. A model is any function $h : \mathcal{X} \to \mathcal{Y}$ for making point predictions. The traditional goal of conformal prediction in the "batch" setting is to take a small *calibration dataset* consisting of labeled examples sampled from $\mathcal{D}$ and use it to endow an arbitrary model $h : \mathcal{X} \to \mathcal{Y}$ with prediction sets $\mathcal{T}_h(x) \subseteq Y$ that have the property that these prediction sets cover the true label with probability $1 - \alpha$ *marginally* for some target miscoverage rate $\alpha$: $\Pr_{(x,y) \sim \mathcal{D}}[y \in \mathcal{T}_h(x)] = 1 - \alpha$. This is a *marginal* coverage guarantee because the probability is taken over the randomness of both $x$ and $y$, without conditioning on anything. In the batch setting (unlike in the sequential setting), labels are not available when the prediction sets are deployed. Our goal in this paper is to give simple, practical algorithms in the batch setting that can give stronger than marginal guarantees — the kinds of *multivalid* guarantees introduced by Gupta et al. (2022); Bastani et al. (2022) in the sequential prediction setting.

Following the literature on conformal prediction (Shafer and Vovk, 2008), our prediction sets are parameterized by an arbitrary *non-conformity score* $s_h : \mathcal{Z} \to \mathbb{R}$ defined as a function of the model $h$. Informally, smaller values of $s_h(x, y)$ should mean that the label $y$ "conforms" more closely to the prediction $h(x)$ made by the model. For example, in a regression setting in which $\mathcal{Y} = \mathbb{R}$, the simplest non-conformity score is $s_h(x, y) = |h(x) - y|$. By now there is a large literature giving more sophisticated non-conformity scores for both regression and classification problems—see Angelopoulos and Bates (2021) for an excellent recent survey. A non-conformity score function $s_h(x, y)$ induces a distribution over non-conformity scores, and if $\tau$ is the $1 - \alpha$ quantile of this distribution (i.e. $\Pr_{(x,y) \sim \mathcal{D}}[s_h(x, y) \leq \tau] = 1 - \alpha$), then defining prediction sets as $\mathcal{T}_h^\tau(x) = \{y : s_h(x, y) \leq \tau\}$ gives $1 - \alpha$ marginal coverage. Split conformal prediction (Papadopoulos et al., 2002; Lei et al., 2018) simply finds a threshold $\tau$ that is an empirical $1 - \alpha$ quantile on the calibration set, and then uses this to deploy the prediction sets $\mathcal{T}_h^\tau(x)$ defined above. Our goal is to give stronger coverage guarantees, and to do so, rather than learning a single threshold $\tau$ from the calibration set,

we will learn a function $f : \mathcal{X} \to \mathbb{R}$ mapping unlabeled examples to thresholds. Such a mapping $f$ induces prediction sets defined as follows: $\mathcal{T}_h^f(x) = \{y : s_h(x, y) \leq f(x)\}$.

Our goal is to find functions $f : \mathcal{X} \to \mathbb{R}$ that give valid conditional coverage guarantees of two sorts. Let $\mathcal{G}$ be an arbitrary collection of *groups*: each group $g \in \mathcal{G}$ is some subset of the feature domain $g \in 2^{\mathcal{X}}$ about which we make no assumption and we write $g(x) = 1$ to denote that $x$ is a member of group $g$. An example $x$ might be a member of multiple groups in $\mathcal{G}$. We want to learn a function $f$ that induces group conditional coverage guarantees—i.e. such that for every $g \in \mathcal{G}$: $\Pr_{(x,y) \sim \mathcal{D}}[y \in T_h^f(x) | g(x) = 1] = 1 - \alpha$. Here we can think of the groups as representing e.g. demographic groups (broken down by race, age, gender, etc) in settings in which we are concerned about fairness, or representing any other categories that we think might be relevant to the domain at hand. Since our functions $f$ now map different examples $x$ to different thresholds $f(x)$, we also want our guarantees to hold conditional on the chosen threshold—which we call a *threshold calibrated* guarantee. This avoids algorithms that achieve their target coverage rates by overcovering for some thresholds and undercovering with others — for example, by randomizing between full and empty prediction sets. That is, we have: $\Pr_{(x,y) \sim \mathcal{D}}[y \in T_h^f(x) | g(x) = 1, f(x) = \tau] = 1 - \alpha$ simultaneously for every $g \in \mathcal{G}$ and every $\tau \in \mathbb{R}$. If $f$ is such that its corresponding prediction sets $T_h^f(x)$ satisfy both group and threshold conditional guarantees simultaneously, then we say that it promises *full multivalid coverage*.

## 1.1 OUR RESULTS

We design, analyze, and empirically evaluate two algorithms: one for giving group conditional guarantees for an arbitrary collection of groups $\mathcal{G}$, and the other for giving full multivalid coverage guarantees for an arbitrary collection of groups $\mathcal{G}$. We give PAC-style guarantees (Park et al., 2019), which means that with high probability over the draw of the calibration set, our deployed prediction sets have their desired coverage properties on the underlying distribution. Thus our algorithms also offer "training-conditional coverage" in the sense of Bian and Barber (2022). We prove our generalization theorems under the assumption that the data is drawn i.i.d. from some distribution, but note that De Finetti's theorem (Ressel, 1985) implies that our analysis carries over to data drawn from any exchangeable distribution (see Remark C.1).

**Group Conditional Coverage: `BatchGCP`** We first give an exceedingly simple algorithm `BatchGCP` (Algorithm 1) to find a model $f$ that produces prediction sets $\mathcal{T}_h^f$ that have group conditional (but not threshold calibrated) coverage guarantees. We consider the class of functions $\mathcal{F} = \{f_\lambda : \lambda \in \mathbb{R}^{|\mathcal{G}|}\}$: each $f_\lambda \in \mathcal{F}$ is parameterized by a vector $\lambda \in \mathbb{R}^{|\mathcal{G}|}$, and takes value: $f_\lambda(x) = f_0(x) + \sum_{g \in \mathcal{G}: g(x)=1} \lambda_g$. Here $f_0$ is some arbitrary initial model. Our algorithm simply finds the parameters $\lambda$ that minimize the *pinball* loss of $f_\lambda(x)$. This is a $|\mathcal{G}|$-dimensional convex optimization problem and so can be solved efficiently using off the shelf convex optimization methods. We prove that the resulting function $f_\lambda(x)$ guarantees group conditional coverage. This can be viewed as an extension of conformalized quantile regression (Romano et al., 2019) which is also based on minimizing pinball loss. It can also be viewed as an algorithm promising "quantile multiaccuracy", by analogy to (mean) multiaccuracy introduced in Hébert-Johnson et al. (2018); Kim et al. (2019), and is related to similar algorithms for guaranteeing multiaccuracy (Gopalan et al., 2022b). Here pinball loss takes the role that squared loss does in (mean) multiaccuracy.

**Multivalid Coverage: `BatchMVP`** We next give a simple iterative algorithm `BatchMVP` (Algorithm 2) to find a model $f$ that produces prediction sets $\mathcal{T}_h^f$ that satisfy both group and threshold conditional guarantees simultaneously — i.e. full multivalid guarantees. It iteratively finds groups $g \in \mathcal{G}$ and thresholds $\tau$ such that the current model fails to have the target coverage guarantees conditional on $g(x) = 1$ and $f(x) = \tau$, and then "patches" the model so that it does. We show that each patch improves the pinball loss of the model substantially, which implies fast convergence. This can be viewed as an algorithm for promising "quantile multicalibration" and is an extension of related algorithms for guaranteeing mean multicalibration (Hébert-Johnson et al., 2018), which offer similar guarantees for mean (rather than quantile) prediction. Once again, pinball loss takes the role that squared loss takes in the analysis of mean multicalibration.

**Empirical Evaluation**   We implement both algorithms, and evaluate them on synthetic prediction tasks, and on 10 real datasets derived from US Census data from the 10 largest US States using the "Folktables" package of Ding et al. (2021). In our synthetic experiments, we measure group conditional coverage with respect to synthetically defined groups that are constructed to be correlated with label noise. On Census datasets, we aim to ensure coverage on population groups defined by reported race and gender categories. We compare our algorithms to two other split conformal prediction methods: a naive baseline which simply ignores group membership, and uses a single threshold, and the method of Foygel Barber et al. (2020), which calibrates a threshold $\tau_g$ for each group $g \in \mathcal{G}$, and then on a new example $x$, predicts the most conservative threshold among all groups $x$ belongs to. We find that both of our methods obtain significantly better group-wise coverage and threshold calibration than the baselines we compare to. Furthermore, both methods are very fast in practice, only taking a few seconds to calibrate on datasets containing tens of thousands of points.

We have discussed the most closely related work; but see Appendix A for an extended discussion of additional related work.

## 2 PRELIMINARIES

We study prediction tasks over a domain $\mathcal{Z} = \mathcal{X} \times \mathcal{Y}$. $\mathcal{X}$ denotes the feature domain and $\mathcal{Y}$ the label domain. We write $\mathcal{G} \subseteq 2^{\mathcal{X}}$ to denote a collection of subsets of $\mathcal{X}$ which we represent as indicator functions $g : \mathcal{X} \to \{0, 1\}$. The label domain might e.g. be real valued ($\mathcal{Y} = \mathbb{R}$)—the *regression* setting, or consist of some finite unordered set—the *multiclass classification* setting.

Suppose there is a fixed distribution $\mathcal{D} \in \Delta \mathcal{Z}$. Given such a distribution, we will write $\mathcal{D}_{\mathcal{X}}$ to denote the marginal distribution over features: $\mathcal{D}_{\mathcal{X}} \in \Delta \mathcal{X}$ induced by $\mathcal{D}$. We will write $\mathcal{D}_{\mathcal{Y}}(x) \in \Delta \mathcal{Y}$ to denote the conditional distribution over labels induced by $\mathcal{D}$ when we condition on a particular feature vector $x$. We sometimes overload the notation and write $\mathcal{D}(x) = \mathcal{D}_{\mathcal{Y}}(x)$.

Our uncertainty quantification is based on a bounded non-conformity score function $s : \mathcal{X} \times \mathcal{Y} \to \mathbb{R}$. Non-conformity score functions are generally defined with respect to some model $h$—which is why in the introduction we wrote $s_h$—but our development will be entirely agnostic to the specifics of the non-conformity score function, and so we will just write $s$ for simplicity. Without loss of generality, we assume that the scoring function takes values in the unit interval: $s(x, y) \in [0, 1]$ for any $x \in \mathcal{X}$ and $y \in \mathcal{Y}$. Given a distribution $\mathcal{D}$ over $\mathcal{Z} = \mathcal{X} \times \mathcal{Y}$ and a non-conformity score function $s$, we write $\mathcal{S}$ to denote the induced joint distribution over feature vectors $x$ and corresponding non-conformity scores $s(x, y)$. Analogously to our $\mathcal{D}(x)$ notation, we write $\mathcal{S}(x)$ to denote the non-conformity score distribution conditional on a particular feature vector $x$. Given a subset of the feature space $B \subset \mathcal{X}$, we write $\mathcal{S}|B$ to denote the conditional distribution on non-conformity scores conditional on $x \in B$. Finally, we assume that all non-conformity score distributions $\mathcal{S}(x)$ are continuous, which simplifies our treatment of quantiles — note that if this is not the case already, it can be enforced by perturbing non-conformity scores with arbitrarily small amounts of noise from a continuous distribution.

**Definition 2.1.** *For any $q \in [0, 1]$, we say that $\tau$ is a $q$-quantile of a (continuous) nonconformity score distribution $\mathcal{S}$ if $\Pr_{(x,s)\sim\mathcal{S}}[s \leq \tau] = q$.*

Our convergence results will be parameterized by the *Lipschitz* parameter of the CDF of the underlying nonconformity score distribution. Informally speaking, a distribution with a Lipschitz CDF cannot concentrate too much of its mass on an interval of tiny width. Similar assumptions are commonly needed in related work, and can be guaranteed by perturbing non-conformity scores with noise — see e.g. the discussion in (Gupta et al., 2022; Bastani et al., 2022).

**Definition 2.2.** *A conditional nonconformity score distribution $\mathcal{S}(x)$ is $\rho$-Lipschitz if we have $\Pr_{s\sim\mathcal{S}(x)}[s \leq \tau'] - \Pr_{s\sim\mathcal{S}(x)}[s \leq \tau] \leq \rho(\tau' - \tau)$ for all $0 \leq \tau \leq \tau' \leq 1$. A nonconformity score distribution $\mathcal{S}$ is $\rho$-Lipschitz if for each $x \in \mathcal{X}$, $\mathcal{S}(x)$ is $\rho$-Lipschitz.*

If we could find a model $f : \mathcal{X} \to [0, 1]$ that on each input $x$ outputs a value $f(x)$ that is a $q$-quantile of the nonconformity score distribution $\mathcal{S}(x)$, this would guarantee true conditional coverage at rate $q$: for every $x$, $\Pr_y[y \in \mathcal{T}^f(x)|x] = q$, where $\mathcal{T}^f(x) = \{y : s(x, y) \leq f(x)\}$. As this is generally impossible, our aim will be to train models $f$ that allow us to make similar claims — not conditional on every $x$, but conditional on membership of $x$ in some group $g$ and on the value of $f(x)$. To facilitate learning models $f$ with guarantees conditional on their output values, we will learn models $f$ whose range $R(f) = \{f(x) : x \in \mathcal{X}\}$ has finite cardinality $m = |R(f)| < \infty$.

## 3 ALGORITHMS

### 3.1 ALGORITHMIC PRELIMINARIES

In this section we establish lemmas that will be key to the analysis of both of the algorithms we give. Both of our algorithms will rely on analyzing *pinball loss* as a potential function.

**Definition 3.1.** *The* pinball loss *at target quantile $q$ for threshold $\tau$ and nonconformity score $s$ is*

$$L_q(\tau, s) = (s - \tau)q \cdot 1[s > \tau] + (\tau - s)(1 - q) \cdot 1[s \leq \tau].$$

*We write $PB_q^{\mathcal{S}}(f) = \underset{(x,s)\sim\mathcal{S}}{\mathbb{E}}[L_q(f(x), s)]$ for a model $f : \mathcal{X} \to [0, 1]$ and nonconformity score distribution $\mathcal{S}$. When quantile $q$ and/or distribution $\mathcal{S}$ is clear, we write $PB$, $PB_q$, and/or $PB^{\mathcal{S}}$.*

It is well known that pinball loss is minimized by the function that predicts for each $x$ the target quantile $q$ of a conditional score distribution given $x$, but we will need a more refined set of statements. First we define a model's *marginal* quantile consistency error:

**Definition 3.2.** *A $q$-quantile predictor $f : \mathcal{X} \to \mathbb{R}$ has marginal quantile consistency error $\alpha$ if $\left|\Pr_{(x,s)\sim\mathcal{S}}[s \leq f(x)] - q\right| = \alpha$. If $\alpha = 0$, we say that $f$ satisfies marginal $q$-quantile consistency.*

Informally, we will need the pinball loss to have both a progress and an "anti-progress" property: If a model $f$ is *far* from satisfying marginal quantile consistency, then linearly shifting it so that it becomes marginal quantile consistent should reduce pinball loss substantially. Conversely, if it *is* marginal quantile consistent, then perturbing it slightly should not increase pinball loss substantially. The next lemma establishes these, under the assumption that the underlying distribution is Lipschitz.

**Lemma 3.1.** *Fix any nonconformity score distribution $\mathcal{S}$ that is $\rho$-Lipschitz. Fix any model $f : \mathcal{X} \to \mathbb{R}$ that has marginal quantile consistency error $\alpha$ with respect to target quantile $q$, and let $f'(x) = f(x) + \Delta$ with $\Delta$ chosen such that $f'$ is marginal quantile consistent at quantile $q$. Then*

$$-\alpha|\Delta| + \frac{\alpha^2}{2\rho} \leq PB_q^{\mathcal{S}}(f') - PB_q^{\mathcal{S}}(f) \leq -\frac{\alpha^2}{2\rho}.$$

We now define what will be a basic building block of our iterative algorithm for obtaining multivalid coverage, and of the *analysis* of our algorithm for obtaining group conditional coverage. It is a simple "patch" operation on a model, that shifts the model's predictions by a constant term $\Delta$ only on those examples $x$ that lie in some subset $B$ of the feature space:

**Definition 3.3** (Patch Operation). *Given a model $f$, a subset $B \subseteq \mathcal{X}$, and a value $\Delta \in \mathbb{R}$ define the patched model $f' = \text{Patch}(f, B, \Delta)$ to be such that*

$$f'(x) = f(x) + \Delta \quad \text{if } x \in B, \quad \text{and } f'(x) = f(x) \text{ otherwise}.$$

We next show that if we have a model $f$, and we can identify a large region $B$ on which it is far from satisfying marginal quantile consistency, then "patching" the model that it satisfies marginal quantile consistency on $S|B$ substantially improves its pinball loss.

**Lemma 3.2.** *Given some predictor $f : \mathcal{X} \to \mathbb{R}$, suppose we have a set of points $B \subseteq \mathcal{X}$ with*

$$\Pr_{(x,s)\sim\mathcal{S}}[x \in B] \cdot \left(\Pr_{(x,s)\sim\mathcal{S}}[s \leq f(x)|x \in B] - q\right)^2 \geq \alpha \quad \text{and} \quad \Delta = \underset{\Delta'\in\mathbb{R}}{\text{argmin}}\left|\Pr_{(x,s)\sim\mathcal{S}}[s \leq f(x) + \Delta'|x \in B] - q\right|.$$

*Then, if $\mathcal{S}|B$ is continuous and $\rho$-Lipschitz, $f' = \text{Patch}(f, B, \Delta)$ has $PB_q^{\mathcal{S}}(f') \leq PB_q^{\mathcal{S}}(f) - \frac{\alpha}{2\rho}$.*

### 3.2 BATCHGCP: OBTAINING GROUP CONDITIONAL GUARANTEES

We now give an extremely simple algorithm `BatchGCP` (Batch Group-Conditional Predictor) that obtains group conditional (but not threshold calibrated) prediction sets. `BatchGCP` only needs to solve a single closed-form convex optimization problem. Specifically, Algorithm 1 takes as input an arbitrary threshold model $f$ and collection of groups $\mathcal{G}$, and then simply minimizes pinball loss over all linear combinations of $f$ and the group indicator functions $g \in \mathcal{G}$. This is a quantile-analogue of a similar algorithm that obtains group conditional *mean* consistency (Gopalan et al., 2022b).

---

**Algorithm 1:** `BatchGCP(f, G, q, D)`

Let $\lambda^*$ be a solution to the optimization problem:

$$\text{argmin}_\lambda \mathbb{E}_{(x,y)\sim\mathcal{D}} \left[ L_q\left( \hat{f}(x;\lambda), s(x,y) \right) \right] \quad \text{where} \quad \hat{f}(x;\lambda) \equiv f(x) + \sum_{g\in\mathcal{G}} \lambda_g \cdot g(x)$$

Output $\hat{f}(x;\lambda^*)$

---

**Theorem 3.1.** *For any input model $f$, groups $\mathcal{G}$, and $q \in [0,1]$, Algorithm 1 returns $\hat{f}(x;\lambda^*)$ with*

$$\Pr_{(x,y)\sim\mathcal{D}}\left[ y \in \mathcal{T}^{\hat{f}}(x) | g(x) = 1 \right] = q \quad \text{for every } g \in \mathcal{G},$$

*where $\mathcal{T}^{\hat{f}}(x) = \{y : s(x,y) \le \hat{f}(x;\lambda^*)\}$. Furthermore, $PB_q^{\mathcal{S}}(\hat{f}(\cdot;\lambda^*)) \le PB_q^{\mathcal{S}}(f)$.*

The analysis is simple: if the optimal model $\hat{f}(x,\lambda^*)$ did not satisfy marginal quantile consistency on some $g \in \mathcal{G}$, then by Lemma 3.2, a patch operation on the set $B(g) = \{x : g(x) = 1\}$ would further reduce its pinball loss. By definition, this patch would just shift the model parameter $\lambda_g^*$ and yield a model $\hat{f}(x,\lambda')$ for $\lambda' \ne \lambda^*$, falsifying the optimality of $\hat{f}(x,\lambda^*)$ among such models.

### 3.3 BATCHMVP: OBTAINING FULL MULTIVALID COVERAGE

In this section, we give a simple iterative algorithm `BatchMVP` (Batch Multivalid Predictor) that trains a threshold model $f$ that provides full multivalid coverage — i.e. that produces prediction sets $\mathcal{T}^f(x)$ that are both group conditionally valid and threshold calibrated. To do this, we start by defining quantile multicalibration, a quantile prediction analogue of (mean) multicalibration defined in Hébert-Johnson et al. (2018); Gopalan et al. (2022a) .

**Definition 3.4.** *The* quantile calibration error *of $q$-quantile predictor $f : \mathcal{X} \to [0,1]$ on group $g$ is:*

$$Q(f,g) = \sum_{v \in R(f)} \Pr_{(x,s)\sim\mathcal{S}}[f(x) = v | g(x) = 1]\Big( q - \Pr_{(x,s)\sim\mathcal{S}}[s \le f(x) | f(x) = v, g(x) = 1] \Big)^2.$$

*We say that $f$ is $\alpha$-approximately $q$-quantile multicalibrated with respect to group collection $\mathcal{G}$ if*

$$Q(f,g) \le \frac{\alpha}{\Pr_{(x,s)\sim\mathcal{S}}[g(x) = 1]} \quad \text{for every } g \in \mathcal{G}.$$

Conditional on membership in each group $g$, the quantile multicalibration error gives a bound on the expected coverage error when conditioning both on membership in $g$ and on a predicted threshold of $f(x) = v$, in expectation over $v$. In particular, it implies (and is stronger than) the following simple worst-case (over $g$ and $v$) bound on coverage error conditional on both $g(x) = 1$ and $f(x) = v$:

**Claim 3.1.** *If $f$ is $\alpha$-approximately quantile multicalibrated with respect to $\mathcal{G}$ and $q$, then*

$$\left| \Pr_{(x,y)\sim\mathcal{D}}\left[ y \in \mathcal{T}^f(x) | g(x) = 1, f(x) = v \right] - q \right| \le \frac{\sqrt{\alpha}}{\sqrt{\Pr_{(x,s)\sim\mathcal{S}}[g(x) = 1, f(x) = v]}} \text{ for } g \in \mathcal{G}, v \in R(f).$$

---

**Algorithm 2:** `BatchMVP(f, α, q, G, ρ, S, m)`

**Initialize** $t = 0$, and define $f_0$ as $f_0(x) = \min_{v\in[\frac{1}{m}]} |v - f(x)|$ for $x \in \mathcal{X}$.

**while** $f_t$ is not $\alpha$-approximately $q$-quantile multicalibrated with respect to $\mathcal{G}$ **do**

    Let $B_t = \{x : f_t(x) = v_t, g_t(x) = 1\}$, **where:**

$$(v_t, g_t) \in \underset{(v,g)\in[\frac{1}{m}]\times\mathcal{G}}{\text{argmax}} \Pr_{(x,s)\sim\mathcal{S}}[f_t(x) = v, g(x) = 1] \left( q - \Pr_{(x,s)\sim\mathcal{S}}[s \le f_t(x) | f_t(x) = v, g(x) = 1] \right)^2$$

    **Let:**
$$\Delta_t = \underset{\Delta\in[\frac{1}{m}]}{\text{argmin}} \left| \Pr_{(x,s)\sim\mathcal{S}}[s \le f_t(x) + \Delta | x \in B_t] - q \right|$$

    **Update** $f_{t+1} \leftarrow \text{Patch}(f_t, B_t, \Delta_t)$ **and** $t \leftarrow t + 1$.

**Output** $f_t$.

---

Algorithm 2 is simple: Given an initial threshold model $f$, collection of groups $\mathcal{G}$, and target quantile $q$, it repeatedly checks whether its current model $f_t$ satisfies $\alpha$-approximate quantile multicalibration. If not, it finds a group $g_t$ and a threshold $v_t$ such that $f_t$ predicts a substantially incorrect quantile conditional on both membership of $x$ in $g_t$ and on $f_t(x) = v_t$ — such a pair is guaranteed to exist if $f_t$ is not approximately quantile multicalibrated. It then fixes this inconsistency and produces a new model $f_{t+1}$ with a patch operation up to a target discretization parameter $m$ which ensures that the range of $f_{t+1}$ does not grow too large. Under the assumption that the non-conformity score distribution is Lipschitz, each patch operation substantially reduces the pinball loss of the current predictor, which ensures fast convergence to quantile multicalibration.

**Theorem 3.2.** *Suppose $\mathcal{S}$ is $\rho$-Lipschitz and continuous, and $m = \frac{8\rho^2}{\alpha}$. After $T \leq \frac{32\rho^3}{\alpha^2}$ many rounds, Algorithm 2* `BatchMVP`$(f, \alpha, q, \mathcal{G}, \rho, \mathcal{S}, m)$ *returns $f_T$ such that*

1. $PB_q^{\mathcal{S}}(f_T) \leq PB_q^{\mathcal{S}}(f_0) - T\frac{\alpha^2}{32\rho^3}$.

2. $f_T$ *is $\alpha$-approximately quantile multicalibrated with respect to $\mathcal{G}$ and $q$. In particular, via Claim 3.1, we have for every $g \in \mathcal{G}$ and $v \in R(f)$,*

$$\left| \Pr_{(x,y)\sim\mathcal{D}} \left[ y \in \mathcal{T}^{f_T}(x) | g(x) = 1, f_T(x) = v \right] - q \right| \leq \frac{\sqrt{\alpha}}{\sqrt{\Pr_{(x,s)\sim\mathcal{S}}[g(x) = 1, f_T(x) = v]}}$$

## 4 OUT OF SAMPLE GENERALIZATION

We have presented `BatchGCP` (Algorithm 1) and `BatchMVP` (2) as if they have direct access to the true data distribution $\mathcal{D}$. In practice, rather than being able to directly access the distribution, we only have a finite *calibration set* $D = (x_i, y_i)_{i=1}^n$ of $n$ data points sampled iid. from $\mathcal{D}$. In this section, we show that if we run our algorithms on the empirical distribution over the sample $D = (x_i, y_i)_{i=1}^n$, then their guarantees hold not only for the empirical distribution over $D$ but also— with high probability — for the true distribution $\mathcal{D}$. We include the generalization guarantees for `BatchGCP` in Appendix C.3 and focus on generalization guarantees for `BatchMVP`.

At a high level, our argument proceeds as follows (although there are a number of technical complications). For any *fixed* model $f$, by standard concentration arguments its in- and out-of-sample quantile calibration error will be close with high probability for a large enough sample size. Our goal is to apply concentration bounds uniformly to *every* model $f_T$ that might be output by our algorithm, and then union bound over all of them to get a high probability bound for whichever model happens to be output. Towards this, we show how to bound the total number of distinct models that can be output as a function of $T$, the total number of iterations of the algorithm. This is possible because at each round $t$, the algorithm performs a patch operation parameterized by a group $g_t \in \mathcal{G}$ (where $|\mathcal{G}| < \infty$), a value $v_t \in [1/m]$, and an update value $\Delta_t \in [1/m]$ — and thus only a finite number of models $f_{t+1}$ can result from patching the current model $f_t$. The difficulty is that our convergence analysis in Theorem 3.2 gives a convergence guarantee in terms of the smoothness parameter $\rho$ of the underlying distribution, which will not be preserved on the *empirical distribution* over the sample $D$ drawn from $\mathcal{D}$. Hence, to upper bound the number of rounds our algorithm can run for, we need to interleave our generalization theorem with our convergence theorem, arguing that at each step—taken with respect to the empirical distribution over $D$—we make progress that can be bounded by the smoothness parameter $\rho$ of the underlying distribution $\mathcal{D}$.

We first prove a high probability generalization bound for our algorithm as a function of the number of steps $T$ it converges in. This bound holds uniformly for all $T$, and so can be applied as a function of the actual (empirical) convergence time $T$.

We let $D = (x_i, y_i)_{i=1}^n$ denote our sample, $S = (x_i, s(x_i, y_i))_{i=1}^n$ our nonconformity score sample, and $\tilde{\mathcal{S}}_S$ denote the empirical distribution over $S$. When $S$ is clear from the context, we just write $\tilde{\mathcal{S}}$.

**Theorem 4.1.** *Suppose $\mathcal{S}$ is $\rho$-Lipschitz and $S \sim \mathcal{S}^n$. Suppose* `BatchMVP`$(f, \alpha, q, \mathcal{G}, \rho, \tilde{\mathcal{S}}_S, m)$ *(Algorithm 2) runs for $T$ rounds and outputs model $f_T$. Then $f_T$ is $\alpha'$-approximately $q$-quantile multicalibrated with respect to $\mathcal{G}$ on $\mathcal{S}$ with probability $1 - \delta$, where*

$$\alpha' = \alpha + 21\sqrt{\frac{3\rho^2\left(\ln(\frac{4\pi^2 T^2}{3\delta}) + T\ln(\frac{\rho^4|\mathcal{G}|}{\alpha^2})\right)}{2\alpha n}} + \frac{12\rho^2(\frac{4\pi^2 T^2}{3\delta}) + T\ln(\frac{\rho^4|\mathcal{G}|}{\alpha^2}))}{\alpha n}.$$

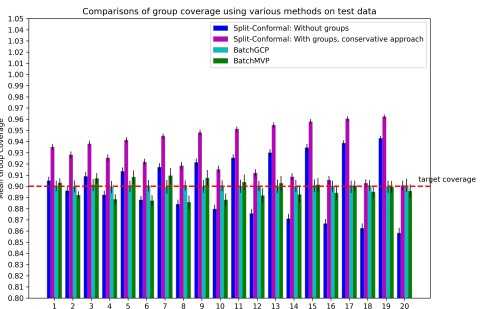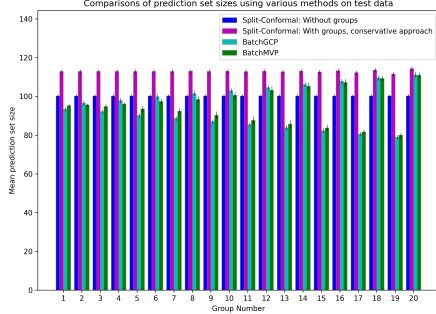

Figure 1: Performance comparisons across different conformal prediction methods. Groupwise coverage is on the left, and mean prediction-set size by group is on the right (averaged over 50 runs). Error bars show standard deviation.

We then prove a worst-case upper bound on the convergence time $T$, which establishes a worst-case PAC style generalization bound in combination with Theorem 4.1. We remark that although the theorem contains a large constant, in our experiments, our algorithm halts after a small number of iterations $T$ (See Sections 5 and D), and we can directly apply Theorem 4.1 with this empirical value for $T$.

**Theorem 4.2.** *Suppose $\mathcal{S}$ is $\rho$-Lipschitz and continuous, $m = \frac{\rho^2}{2\alpha}$, and our calibration set $S \sim \mathcal{S}^n$ consists of $n$ iid. samples drawn from $\mathcal{S}$, where $n$ is sufficiently large: $n \geq 92928 \left( \ln \left( \frac{128\rho^3}{\alpha^2\delta} \right) + \frac{8\rho^3}{\alpha^2} \ln \left( \frac{\rho^4 |\mathcal{G}|}{\alpha^2} \right) \right) \max \left( \frac{\rho^4}{4\alpha^4}, \frac{\rho^6}{\alpha^4} \right)$. Then $\mathtt{BatchMVP}(f, \alpha, q, \mathcal{G}, \rho, \bar{\mathcal{S}}_S, m)$ (Algorithm 2) halts after $T \leq \frac{8\rho^3}{\alpha^2}$ rounds with prob. $1 - \delta$.*

Formal generalization arguments are in Appendix C. Theorems 4.1, 4.2 are proved in Appendix C.4.

## 5 EXPERIMENTS

### 5.1 A SYNTHETIC REGRESSION TASK

We first consider a linear regression problem on synthetic data drawn from a feature domain of 10 binary features and 90 continuous features, with each binary feature drawn uniformly at random, and each continuous feature drawn from a normal distribution $\mathcal{N}(0, \sigma_x^2)$. The 10 binary features are used to define 20 intersecting groups, each depending on the value of a single binary feature. An input's label is specified by an ordinary least squares model with group-dependent noise as: $y = \langle \theta, x \rangle + \mathcal{N} \left( 0, \sigma^2 + \sum_{i=1}^{10} \sigma_i^2 x_i \right)$, where each term $\sigma_i^2$ is associated with one binary feature. We set $\sigma_i^2 = i$ for all $i \in [10]$. So the more groups an input $x$ is a member of, the more label noise it has, with larger index groups contributing larger amounts of label noise.

We generate a dataset $\{(x_i, y_i)\}$ of size 40000 using the above-described model, and split it evenly into training and test data. The training data is further split into training data of size 5000 (with which to train a least squares regression model $f$) and calibration data of size 15000 (with which to calibrate our various conformal prediction methods). Given the trained predictor $f$, we use non-conformity score $s(x, y) = |f(x) - y|$. Next, we define the set of groups $\mathcal{G} = \{g_1, g_2, \cdots, g_{20}\}$ where for each $j \in [20]$, $g_j = \{x \in \mathcal{X} \mid x_{\lfloor (j+1)/2 \rfloor} \equiv_2 j + 1\}$. We run Algorithm 1 ($\mathtt{BatchGCP}$) and Algorithm 2 ($\mathtt{BatchMVP}$ with $m = 100$ buckets) to achieve group-conditional and full multivalid coverage guarantees respectively, with respect to $\mathcal{G}$ with target coverage $q = 0.9$. We compare the performance of these methods to naive split-conformal prediction (which, without knowledge of $\mathcal{G}$, uses the calibration data to predict a single threshold) and the method of Foygel Barber et al. (2020) which predicts a threshold for each group in $\mathcal{G}$ that an input $x$ is part of, and chooses the most conservative one.

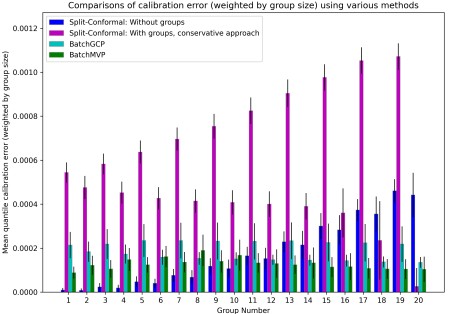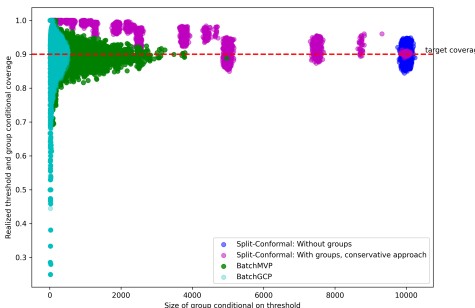

Figure 2: The left figure plots group-wise calibration error (averaged over 50 runs), weighted by group size. Error bars show standard deviation. The right figure is a scatterplot of the number of points associated with each threshold-group pair $(g, \tau_i)$ against the average coverage conditional on that pair for all $g \in \mathcal{G}$ and all $\tau_i$ in a grid, over all tested conformal prediction methods (consolidating results over all 50 runs). Target coverage is $q = 0.9$.

Figures 1 and 2 present results over 50 runs of this experiment. Notice that both `BatchGCP` and `BatchMVP` achieve close to the desired group coverage across all groups—with `BatchGCP` achieving nearly perfect coverage and `BatchMVP` sometimes deviating from the target coverage rate by as much as 1%. In contrast, the method of Foygel Barber et al. (2020) significantly overcovers on nearly all groups, particularly low-noise groups, and naive split-conformal starts undercovering and overcovering on high-noise and low-noise groups respectively as the expected label-noise increases. Group-wise calibration error is high across all groups but the last using the method of Foygel Barber et al. (2020), and naive split-conformal has low calibration error on groups where inclusion/exclusion reflects less fluctuation in noise, and higher calibration error in groups where there is much higher fluctuation in noise based on inclusion. Both `BatchGCP` and `BatchMVP` have lower group calibration errors — interestingly, `BatchGCP` appears to do nearly as well as `BatchMVP` in this regard despite having no theoretical guarantee on threshold calibration error. A quantile multicalibrated predictor must have low coverage error conditional on groups $g$ and thresholds $\tau_i$ that appear frequently, and may have high coverage error for pairs that appear infrequently—behavior that we see in `BatchMVP` in Figure 2. On the other hand, for both naive split conformal prediction and the method of Foygel Barber et al. (2020), we see high mis-coverage error even for pairs $(g, \tau_i)$ containing a large fraction of the probability mass.

We fix the upper-limit of allowed iterations $T$ for `BatchMVP` to 1000, but typically the algorithm converges and halts in many fewer iterations. Across the 50 runs of this experiment, the average number of iterations $T$ taken to converge was $45.54 \pm 14.77$.

## 5.2 AN INCOME PREDICTION TASK ON CENSUS DATA

We also compare our methods to naive split conformal prediction and the method of Foygel Barber et al. (2020) on real data from the 2018 Census American Community Survey Public Use Microdata, compiled by the Folktables package (Ding et al., 2021). This dataset records information about individuals including race, gender and income. In this experiment, we generate prediction sets for a person's income while aiming for valid coverage on intersecting groups defined by race and gender.

The Folktables package provides datasets for all 50 US states. We run experiments on state-wide datasets: for each state, we split it into 60% training data $\mathcal{D}_{train}$ for the income-predictor, 20% calibration data $\mathcal{D}_{calib}$ to calibrate the conformal predictors, and 20% test data $\mathcal{D}_{test}$. After training the income-predictor $f$ on $\mathcal{D}_{train}$, we use the non-conformity score $s(x, y) = |f(x) - y|$. There are 9 provided codes for race[1] and 2 codes for sex (1. Male, 2. Female) in the Folktables data. We

---

[1]1. White alone, 2. Black or African American alone, 3. American Indian alone, 4. Alaska Native alone, 5. American Indian and Alaska Native tribes specified; or American Indian or Alaska Native, not specified and no other races, 6. Asian alone, 7. Native Hawaiian and other Pacific Islander alone, 8. Some Other Race alone, 9. Two or More Races.

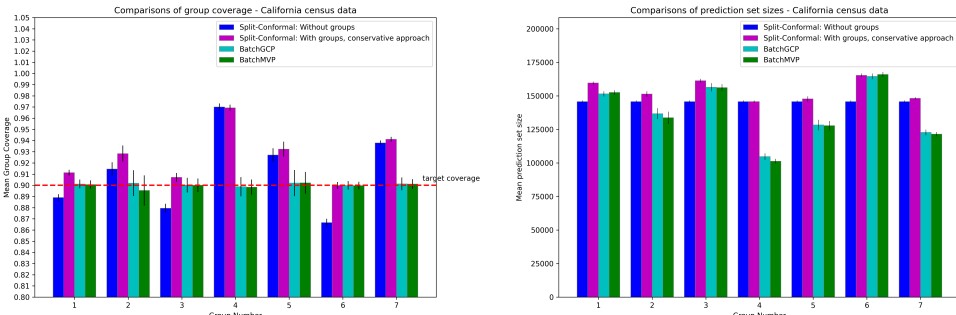

Figure 3: Performance across different conformal prediction methods on Folktables California data. Groupwise coverage is on the left, and mean prediction-set sizes for each group are on the right (averaged over 50 rounds). Error bars show standard deviation.

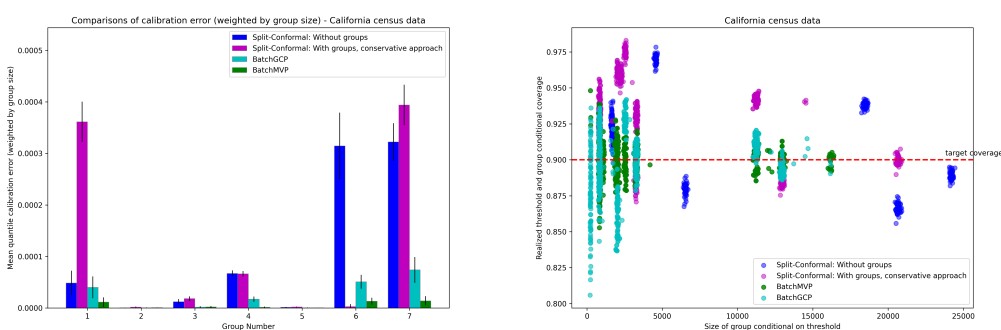

Figure 4: The left figure plots group-wise calibration error, weighted by group size (averaged over 50 runs). Error bars show standard deviation. The right figure is a scatterplot of the number of points associated with each threshold-group pair $(g, \tau_i)$ against the average coverage conditional on that pair for all $g \in \mathcal{G}$ and all $\tau_i$ in a grid, over all tested conformal prediction methods (consolidating results over all 50 runs). Target coverage is $q = 0.9$.

define groups for five out of nine race codes (disregarding the four with the least amount of data) and both sex codes. We run all four algorithms (naive split-conformal, the method of Foygel Barber et al. (2020), `BatchGCP`, and `BatchMVP` with $m = 300$ buckets) with target coverage $q = 0.9$.

We ran this experiment on the 10 largest states. Figure 3 and Figure 4 present comparisons of the performance of all four algorithms on data taken from California, averaged over 50 different runs. Results on all remaining states are presented in Appendix D.2.

Just as in the synthetic experiments, both `BatchGCP` and `BatchMVP` achieve excellent coverage across all groups—in fact now, we see nearly perfect coverage for *both* `BatchGCP` and `BatchMVP`, with `BatchGCP` still obtaining slightly better group conditional coverage. In contrast, naive split-conformal prediction undercovers on certain groups and overcovers on others, and the method of Foygel Barber et al. (2020) significantly overcovers on some groups (here, group 4 and 7). The conservative approach also generally yields prediction sets of much larger size. We see also that `BatchMVP` achieves very low rates of calibration error across all groups, outperforming `BatchGCP` in this regard. Calibration error is quite irregular across groups for both naive split-conformal prediction and for the method of Foygel Barber et al. (2020), being essentially zero in certain groups and comparatively much larger in others. The average number of iterations $T$ `BatchMVP` converged in was $10.64 \pm 1.12$.

**Reproducibility statement** The full Python implementations of `BatchGCP` and `BatchMVP` can be found in the supplementary zip. Jupyter notebooks that implement each of our experiments are

also included in the supplementary zip. For details on our experiments, such as how we generate the data, what conformal scores we use, how we instantiate `BatchMVP` and `BatchGCP`, please see Section 5 and Appendix D, as well as the corresponding Jupyter notebooks. Proofs of all theoretical results are included, either in the main body of the paper or in the designated Appendices.

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

## A  ADDITIONAL RELATED WORK

Conformal prediction (see Shafer and Vovk (2008); Angelopoulos and Bates (2021) for excellent surveys) has seen a surge of activity in recent years. One large category of recent work has been the development of sophisticated non-conformity scores that yield compelling empirical coverage in various settings when paired with split conformal prediction (Papadopoulos et al., 2002; Lei et al., 2018). This includes Romano et al. (2019) who give a nonconformity score based on quantile regression, Angelopoulos et al. (2020) and Romano et al. (2020b) who give nonconformity scores designed for classification, and Hoff (2021) who gives a nonconformity score that leads to Bayes optimal coverage when data is drawn from the assumed prior distribution. This line of work is complementary to ours: we give algorithms that can be used as drop-in replacements for split conformal prediction, and can make use of any of these nonconformity scores.

Another line of work has focused on giving group conditional guarantees. Romano et al. (2020a) note the need for group-conditional guarantees with respect to demographic groups when fairness is a concern, and propose separately calibrating on each group in settings in which the groups are disjoint. Foygel Barber et al. (2020) consider the case of intersecting groups, and give an algorithm that separately calibrates on each group, and then uses the most conservative group-wise threshold when faced with examples that are members of multiple groups — the result is that this method systematically over-covers. The kind of "multivalid" prediction sets that we study here were first proposed by Jung et al. (2021) in the special case of prediction intervals: but the algorithm given by Jung et al. (2021), based on calibrating to moments of the label distribution and using Chebyshev's inequality, also generally leads to over-coverage. Gupta et al. (2022) gave a theoretical derivation of an algorithm to obtain tight multivalid prediction intervals in the sequential adversarial setting, and Bastani et al. (2022) gave a practical algorithm to obtain tight multivalid coverage in the general case — also in the sequential adversarial setting. Although the sequential setting is more difficult in the sense that it makes no distributional assumptions, it also requires that labels be available after

predictions are made at test time, in contrast to the batch setting that we study, in which labels are not available. Gupta et al. (2022) give an online-to-batch reduction that requires running the sequential algorithm on a large calibration set, saving the algorithm's internal state at each iteration, and then deploying a randomized predictor that randomizes over the internal state of the algorithm across all rounds of training. This is generally impractical for large datasets; in contrast we give a direct, simple, deterministic predictor in the batch setting.

Our algorithms can be viewed as learning quantile multiaccurate predictors and quantile multicalibrated predictors respectively — by analogy to multiaccuracy Kim et al. (2019) and multicalibration Hébert-Johnson et al. (2018) which are defined with respect to means instead of quantiles. Their analysis is similar, but with pinball loss playing the role played by squared error in (mean) multicalibration. This requires analyzing the evolution of pinball loss under Lipschitzness assumptions on the underlying distribution, which is a complication that does not arise for means. More generally, our goals for obtaining group conditional guarantees for intersecting groups emerge from the literature on "multigroup" fairness — see e.g. (Kearns et al., 2018; Hébert-Johnson et al., 2018; Kearns et al., 2019; Rothblum and Yona, 2021; Dwork et al., 2019; Globus-Harris et al., 2022)

## B  MISSING DETAILS FROM SECTION 3

### B.1  MISSING DETAILS FROM SECTION 3.1

**Lemma B.1.** *Fix any $x \in \mathcal{X}$ and $\tau \in [0, 1]$.*

$$\mathbb{E}_{s \sim \mathcal{S}(x)}\left[\frac{dL_q(f(x) + \tau, s)}{d\tau}\right] = \int_{[0,1]} \frac{dL_q(f(x) + \tau, s)}{d\tau}d\mathcal{S}(x) = \Pr_{s \sim \mathcal{S}(x)}[s \leq f(x) + \tau] - q$$

*Proof.*

$$\begin{aligned}
\mathbb{E}_{s \sim \mathcal{S}(x)}\left[\frac{dL_q(f(x) + \tau), s}{d\tau}\right] &= \int_0^1 \frac{d}{d\tau}L_q(f(x) + \tau, s)d\mathcal{S}(x) \\
&= \int_0^\tau \frac{d}{d\tau}L_q(f(x) + \tau, s)d\mathcal{S}(x) + \int_\tau^1 \frac{d}{d\tau}L_q(f(x) + \tau, s)d\mathcal{S}(x) \\
&= \int_0^\tau (1 - q)d\mathcal{S}(x) - \int_\tau^1 q d\mathcal{S}(x) \\
&= \Pr_{s \sim \mathcal{S}(x)}[s \leq \tau] - q
\end{aligned}$$

$\square$

**Lemma 3.1.** *Fix any nonconformity score distribution $\mathcal{S}$ that is $\rho$-Lipschitz. Fix any model $f : \mathcal{X} \to \mathbb{R}$ that has marginal quantile consistency error $\alpha$ with respect to target quantile $q$, and let $f'(x) = f(x) + \Delta$ with $\Delta$ chosen such that $f'$ is marginal quantile consistent at quantile $q$. Then*

$$-\alpha|\Delta| + \frac{\alpha^2}{2\rho} \leq PB_q^\mathcal{S}(f') - PB_q^\mathcal{S}(f) \leq -\frac{\alpha^2}{2\rho}.$$

*Proof.*

$$\begin{aligned}
&PB_q^\mathcal{S}(f') - PB_q^\mathcal{S}(f) \\
&= \int_{\mathcal{X} \times [0,1]} L_q(f(x) + \Delta, s) - L_q(f(x), s)\mathcal{S}(dx, ds) \\
&= \int_{\mathcal{X} \times [0,1]} \int_0^\Delta \frac{dL_q(f(x) + \tau, s)}{d\tau}d\tau\mathcal{S}(dx, ds) \\
&= \int_0^\Delta \int_{\mathcal{X} \times [0,1]} \frac{dL_q(f(x) + \tau, s)}{d\tau}\mathcal{S}(dx, ds)d\tau \\
&= \int_0^\Delta \left(\mathbb{E}_{x \sim \mathcal{D}_\mathcal{X}}\left[\Pr_{s \sim \mathcal{S}(x)}[s \leq f(x) + \tau] - q\right]\right)d\tau
\end{aligned}$$

$$= \int_0^\Delta \left( \Pr_{(x,s)\sim\mathcal{S}}[s \le f(x) + \tau] - q \right) d\tau$$

$$= \int_0^\Delta \Pr_{(x,s)\sim\mathcal{S}}[s \le f(x) + \tau]d\tau - \Delta q$$

where the fourth equality follows from Lemma B.1.

For convenience, write $H_{\mathcal{S},f}(\tau) = \Pr_{(x,s)\sim\mathcal{S}}[s \le f(x) + \tau]$. Note that

$$\int_0^\Delta H_{\mathcal{S},f}(\tau)d\tau = \int_0^\Delta \Pr_{(x,s)\sim\mathcal{S}}[s \le f(x) + \tau]d\tau$$

$$= \int_0^\Delta \Pr_{(x,s)\sim\mathcal{S}}[s \le f'(x) - \Delta + \tau]d\tau$$

$$= -\int_0^{-\Delta} \Pr_{(x,s)\sim\mathcal{S}}[s \le f'(x) + \tau]d\tau$$

$$= -\int_0^{-\Delta} H_{\mathcal{S},f'}(\tau)d\tau,$$

as instead of sweeping the area under the curve from $f(x)$ to $f(x)+\Delta$, we can sweep the area under the curve from $f'(x)$ to $f'(x) - \Delta$ because $f'(x) = f(x) + \Delta$.

**Lemma B.2.** *Fix any conformity score distribution $\mathcal{S}$ that is $\rho$-Lipschitz, $\Delta > 0$, and $f : \mathcal{X} \to \mathbb{R}$. Then we have*

$$H_{\mathcal{S},f}(0)\Delta + \frac{(H_{\mathcal{S},f}(\Delta) - H_{\mathcal{S},f}(0))^2}{2\rho}$$

$$\le \int_0^\Delta H_{\mathcal{S},f}(\tau)d\tau$$

$$\le H_{\mathcal{S},f}(\Delta)\Delta - \left( \frac{(H_{\mathcal{S},f}(\Delta) - H_{\mathcal{S},f}(0))^2}{2\rho} \right).$$

*Proof.* For simplicity write $H(\tau) = H_{\mathcal{S},f}(\tau)$. Note that $H(\tau)$ is a non-negative function that is increasing in $\tau$.

First, we find an upper bound the area. The maximum area that can be achieved is when there's a linear rate of increase from $y = H(0)$ to $y = H(\Delta)$ between $x = 0$ and $x = \frac{H(\Delta)-H(0)}{\rho}$ as depicted in Figure 5. The area measured via the integral can be calculated by subtracting the area of the triangle from the area of the rectangle from $x = 0$ to $x = \Delta$ and from $y = 0$ to $y = H(\Delta)$.

$$\int_0^\Delta \Pr_{(x,s)\sim\mathcal{S}}[s \le f(x) + \tau]d\tau$$

$$\le H(\Delta)\Delta - \frac{(H(\Delta) - H(0))^2}{2\rho}.$$

Now, we find a lower bound of the area. The area under the curve is minimized when there's a linear increase from $H(0)$ to $H(\Delta)$ between $x = \Delta - \frac{H(\Delta)-H(0)}{\rho}$ and $x = \Delta$. The area can be calculated as the sum of the area of the rectangle from $x = 0$ to $x = \Delta$ and from $y = 0$ to $y = H(0)$ and the area of the triangle.

$$\int_0^\Delta \Pr_{(x,s)\sim\mathcal{S}}[s \le f(x) + \tau]d\tau$$

$$\ge H(0)\Delta + \frac{H(\Delta) - H(0)}{2\rho}(H(\Delta) - H(0))$$

$\square$

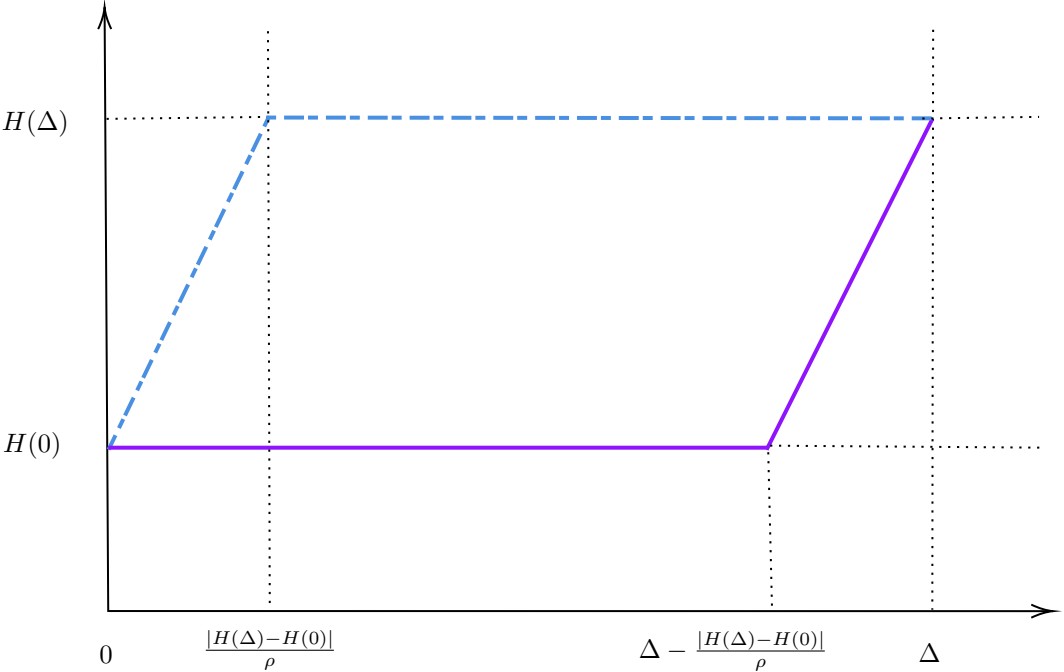

Figure 5: Upper and lower bounding the local area under $H$. The max-area CDF curve is in dashed blue, the min-area CDF curve is in solid purple.

**Case (i)** $\Delta \geq 0$: We have $H_{\mathcal{S},f}(\Delta) = q$ and $H_{\mathcal{S},f}(0) = q - \alpha$.

$$PB_q(f') - PB_q(f) \leq H_{\mathcal{S},f}(\Delta)\Delta - \left(\frac{(H_{\mathcal{S},f}(\Delta) - H_{\mathcal{S},f}(0))^2}{2\rho}\right) - \Delta q$$

$$\leq -\frac{\alpha^2}{2\rho}.$$

On the other hand,

$$PB_q(f') - PB_q(f) \geq H_{\mathcal{S},f}(0)\Delta + \frac{(H_{\mathcal{S},f}(\Delta) - H_{\mathcal{S},f}(0))^2}{2\rho} - \Delta q$$

$$= (q - \alpha)\Delta + \frac{\alpha^2}{2\rho} - \Delta q$$

$$= -\alpha\Delta + \frac{\alpha^2}{2\rho}.$$

**Case (ii)** $\Delta < 0$: We have we have $H_{\mathcal{S},f}(\Delta) = H_{\mathcal{S},f'}(0) = q$ and $H_{\mathcal{S},f}(0) = H_{\mathcal{S},f'}(\Delta') = q + \alpha$ where $\Delta' = -\Delta$, we have

$$PB_q(f') - PB_q(f) = -\int_0^{-\Delta} H_{\mathcal{S},f'}(\tau)d\tau - q\Delta$$

$$= -\int_0^{\Delta'} H_{\mathcal{S},f'}(\tau)d\tau - q\Delta$$

$$\leq -\left(H_{\mathcal{S},f'}(0)\Delta' + \frac{(H_{\mathcal{S},f'}(\Delta) - H_{\mathcal{S},f'}(0))^2}{2\rho}\right) - q\Delta$$

$$\leq -\frac{\alpha^2}{2\rho}.$$

On the other hand, we have

$$
\begin{aligned}
PB_q(f') - PB_q(f) &= -\int_0^{-\Delta} H_{\mathcal{S},f'}(\tau)d\tau - q\Delta \\
&= -\int_0^{\Delta'} H_{\mathcal{S},f'}(\tau)d\tau - q\Delta \\
&\geq -\left(H_{\mathcal{S},f'}(\Delta')\Delta' - \left(\frac{(H_{\mathcal{S},f'}(\Delta') - H_{\mathcal{S},f'}(0))^2}{2\rho}\right)\right) - q\Delta \\
&= (q+\alpha)\Delta + \frac{\alpha^2}{2\rho} \\
&= \alpha\Delta + \frac{\alpha^2}{2\rho}.
\end{aligned}
$$

$\square$

**Lemma 3.2.** *Given some predictor $f : \mathcal{X} \to \mathbb{R}$, suppose we have a set of points $B \subseteq \mathcal{X}$ with*

$$
\Pr_{(x,s)\sim\mathcal{S}}[x \in B] \cdot \left(\Pr_{(x,s)\sim\mathcal{S}}[s \leq f(x)|x \in B] - q\right)^2 \geq \alpha \quad \text{and} \quad \Delta = \operatorname*{argmin}_{\Delta'\in\mathbb{R}}\left|\Pr_{(x,s)\sim\mathcal{S}}[s \leq f(x) + \Delta'|x \in B] - q\right|.
$$

*Then, if $\mathcal{S}|B$ is continuous and $\rho$-Lipschitz, $f' = \mathrm{Patch}(f, B, \Delta)$ has $PB_q^{\mathcal{S}}(f') \leq PB_q^{\mathcal{S}}(f) - \frac{\alpha}{2\rho}$.*

*Proof.* Note that $f$'s marginal quantile consistency error with respect to target quantile $q$ as measured on $\mathcal{S}|B$ is

$$
\left|\Pr_{(x,s)\sim\mathcal{S}|B}[s \leq f(x)] - q\right| \geq \sqrt{\frac{\alpha}{\Pr_{(x,s)\sim\mathcal{S}}[x \in B]}}.
$$

Also, since $\mathcal{S}|B$ is continuous, we have

$$
q = \Pr_{(x,s)\sim\mathcal{S}|B}[s \leq f(x) + \Delta].
$$

In other words, $f'$ satisfies marginal quantile consistency for $q$ as measured on $\mathcal{S}|B$.

Applying Lemma 3.1 to $\mathcal{S}|B$, we have

$$
\begin{aligned}
&PB_q^{\mathcal{S}}(f') - PB_q^{\mathcal{S}}(f) \\
&= \Pr_{(x,s)\sim\mathcal{S}}[x \in B] \cdot (PB_q^{\mathcal{S}|B}(f') - PB_q^{\mathcal{S}|B}(f)) \\
&\leq -\Pr_{(x,s)\sim\mathcal{S}}[x \in B] \cdot \frac{\alpha}{2\rho\Pr_{(x,s)\sim\mathcal{S}}[x \in B]} \\
&= -\frac{\alpha}{2\rho}.
\end{aligned}
$$

$\square$

## B.2   MISSING DETAILS FROM SECTION 3.2

**Theorem 3.1.** *For any input model $f$, groups $\mathcal{G}$, and $q \in [0,1]$, Algorithm 1 returns $\hat{f}(x;\lambda^*)$ with*

$$
\Pr_{(x,y)\sim\mathcal{D}}\left[y \in \mathcal{T}^{\hat{f}}(x)|g(x) = 1\right] = q \quad \text{for every } g \in \mathcal{G},
$$

*where $\mathcal{T}^{\hat{f}}(x) = \{y : s(x,y) \leq \hat{f}(x;\lambda^*)\}$. Furthermore, $PB_q^{\mathcal{S}}(\hat{f}(\cdot;\lambda^*)) \leq PB_q^{\mathcal{S}}(f)$.*

*Proof.* Suppose $\hat{f}(x;\lambda^*)$ is not marginally quantile consistent on some $g' \in \mathcal{G}$ — i.e. $\Pr_{(x,s)}[g'(x) = 1] \cdot (\Pr_{(x,s)\sim\mathcal{S}}[s \leq f(x)|g'(x) = 1] - q)^2 > 0$. In other words, there exists some $\Delta \neq 0$ such that

$$
\Pr_{(x,s)\sim\mathcal{S}}[s \leq f(x) + \Delta|g'(x) = 1] = q.
$$

Suppose we patch $\hat{f}(\cdot; \lambda^*)$

$$f' = \text{Patch}(\hat{f}(\cdot; \lambda^*), g', \Delta)$$

which can be re-written as

$$f'(x) = \hat{f}(x; \lambda^*) + \Delta \cdot \mathbb{1}[g'(x) = 1]$$

$$= f(x) + \left( \sum_{g \in \mathcal{G}} \lambda_g \cdot g(x) \right) + \Delta \cdot \mathbb{1}[g'(x) = 1]$$

$$= f(x) + \sum_{g \in \mathcal{G}} \lambda'_g \cdot g(x)$$

where $\lambda'_{g'} = \lambda^*_{g'} + \Delta$ and $\lambda'_g = \lambda^*_g$ for all other $g \neq g'$.

Lemma 3.2 yields

$$PB_q^{\mathcal{S}}(\hat{f}(\cdot, \lambda')) - PB_q^{\mathcal{S}}(\hat{f}(\cdot; \lambda^*)) < 0$$

which contradicts the fact that $\lambda^*$ is an optimal solution to $\min_\lambda \mathbb{E}_{(x,y) \sim \mathcal{D}} \left[ L_q \left( \hat{f}(x; \lambda), y \right) \right]$. □

## B.3 MISSING DETAILS FROM SECTION 3.3

**Claim 3.1.** *If $f$ is $\alpha$-approximately quantile multicalibrated with respect to $\mathcal{G}$ and $q$, then*

$$\left| \Pr_{(x,y) \sim \mathcal{D}} \left[ y \in \mathcal{T}^f(x) | g(x) = 1, f(x) = v \right] - q \right| \leq \frac{\sqrt{\alpha}}{\sqrt{\Pr_{(x,s) \sim \mathcal{S}}[g(x) = 1, f(x) = v]}} \text{ for } g \in \mathcal{G}, v \in R(f).$$

*Proof.* Consider any $g \in \mathcal{G}$.

$$Q(f, g) = \sum_{v \in R(f)} \Pr_{(x,s) \sim \mathcal{S}}[f(x) = v | g(x) = 1] \left( q - \Pr_{(x,s) \sim \mathcal{S}}[s \leq f(x) | f(x) = v, g(x) = 1] \right)^2$$

$$\leq \frac{\alpha}{\Pr_{(x,s) \sim \mathcal{S}}[g(x) = 1]}$$

In particular, since each term in the sum is non-negative, we must have for every $v \in R(f)$:

$$\Pr_{(x,s) \sim \mathcal{S}}[f(x) = v | g(x) = 1] \left( q - \Pr_{(x,s) \sim \mathcal{S}}[s \leq f(x) | f(x) = v, g(x) = 1] \right)^2 \leq \frac{\alpha}{\Pr_{(x,s) \sim \mathcal{S}}[g(x) = 1]}$$

Dividing both sides by $\Pr_{(x,s) \sim \mathcal{S}}[f(x) = v | g(x) = 1]$ we see that this is equivalent to:

$$\left( q - \Pr_{(x,s) \sim \mathcal{S}}[s \leq f(x) | f(x) = v, g(x) = 1] \right)^2 \leq \frac{\alpha}{\Pr_{(x,s) \sim \mathcal{S}}[g(x) = 1, f_T(x) = v]}$$

Taking the square root yields our claim. □

**Theorem 3.2.** *Suppose $\mathcal{S}$ is $\rho$-Lipschitz and continuous, and $m = \frac{8\rho^2}{\alpha}$. After $T \leq \frac{32\rho^3}{\alpha^2}$ many rounds, Algorithm 2 $\texttt{BatchMVP}(f, \alpha, q, \mathcal{G}, \rho, \mathcal{S}, m)$ returns $f_T$ such that*

1. $PB_q^{\mathcal{S}}(f_T) \leq PB_q^{\mathcal{S}}(f_0) - T\frac{\alpha^2}{32\rho^3}.$

2. *$f_T$ is $\alpha$-approximately quantile multicalibrated with respect to $\mathcal{G}$ and $q$. In particular, via Claim 3.1, we have for every $g \in \mathcal{G}$ and $v \in R(f)$,*

$$\left| \Pr_{(x,y) \sim \mathcal{D}} \left[ y \in \mathcal{T}^{f_T}(x) | g(x) = 1, f_T(x) = v \right] - q \right| \leq \frac{\sqrt{\alpha}}{\sqrt{\Pr_{(x,s) \sim \mathcal{S}}[g(x) = 1, f_T(x) = v]}}$$

*Proof.*

**(1) Marginal quantile consistency error in each round $t$:** At any round $t$, if $f_t$ is not $\alpha$-approximately quantile multicalibrated with respect to $\mathcal{G}$ and $q$, we have

$$\Pr_{(x,s)\sim\mathcal{S}}[x \in B_t]\left(q - \Pr_{(x,s)\sim\mathcal{S}}[s \leq f_t(x)|x \in B_t]\right)^2 \geq \frac{\alpha}{m+1} \geq \frac{\alpha}{2m}:$$

as the average of $m+1$ elements is greater than $\alpha$.

**(2) Decomposing the patch operation into two patch operations:** Write

$$\tilde{\Delta}_t = \operatorname*{argmin}_{\Delta \in [0,1]}\left|\Pr_{(x,s)\sim\mathcal{S}}[s \leq f_t(x) + \Delta|x \in B_t] - q\right|$$

to denote how much we would have patched $f_t$ by if we actually optimized over the unit interval. Then, we can divide the patch operation into two where

$$\tilde{f}_{t+1} = \operatorname{Patch}\left(f_t, B_t, \tilde{\Delta}_t\right)$$

$$f_{t+1} = \operatorname{Patch}\left(\tilde{f}_t, B_t, \Delta_t - \tilde{\Delta}_t\right).$$

Now, we try to bound the change in the pinball loss separately:

$$PB_q^{\mathcal{S}}(f_{t+1}) - PB_q^{\mathcal{S}}(f_t) = \underbrace{(PB_q^{\mathcal{S}}(f_{t+1}) - PB_q^{\mathcal{S}}(\tilde{f}_{t+1}))}_{(*)} + \underbrace{(PB_q^{\mathcal{S}}(\tilde{f}_{t+1}) - PB_q^{\mathcal{S}}(f_t))}_{(**)}.$$

**(2) Bounding (\*\*):** Lemma 3.2 yields

$$(**) = PB^{\mathcal{S}}(\tilde{f}_{t+1}) - PB^{\mathcal{S}}(f_t) \leq -\frac{\alpha}{4\rho m}$$

**(3) Bounding (\*):** Note that $\frac{i}{m} \leq \tilde{\Delta} \leq \frac{i+1}{m}$ for some $i \in [0, \ldots, m-1]$. Because the function $\Delta \to \Pr_{(x,s)\sim\mathcal{S}|B_t}[s \leq f_t(x) + \Delta]$ is an increasing function in $\Delta$, we have

$$\left|\Pr_{(x,s)\sim\mathcal{S}|B_t}[s \leq f_t(x) + \Delta'] - q\right| < \left|\Pr_{(x,s)\sim\mathcal{S}|B_t}\left[s \leq f_t(x) + \frac{i}{m}\right] - q\right| \quad \text{for any } \Delta < \frac{i}{m}$$

$$\left|\Pr_{(x,s)\sim\mathcal{S}|B_t}[s \leq f_t(x) + \Delta'] - q\right| < \left|\Pr_{(x,s)\sim\mathcal{S}|B_t}\left[s \leq f_t(x) + \frac{i+1}{m}\right] - q\right| \quad \text{for any } \Delta > \frac{i+1}{m}.$$

In other words, $\Delta_t \in \{\frac{i}{m}, \frac{i+1}{m}\}$ so we have $|\Delta_t - \tilde{\Delta}_t| \leq \frac{1}{m}$. Because $\mathcal{S}|B_t$ is $\rho$-Lipschitz and $\tilde{f}_{t+1}(x)$ is $q$-quantile for $\mathcal{S}|B_t$, we can bound the marginal quantile consistency error of $f_{t+1}$ against $\mathcal{S}|B_t$ as

$$\left|\Pr_{(x,s)\sim\mathcal{S}|B_t}[s \leq f_{t+1}(x)] - q\right| = \left|\Pr_{(x,s)\sim\mathcal{S}|B_t}[s \leq f_{t+1}(x)] - \Pr_{(x,s)\sim\mathcal{S}|B_t}[s \leq \tilde{f}_t(x) - \tilde{\Delta}_t + \Delta_t]\right| \leq \frac{\rho}{m}$$

as $|\Delta_t - \tilde{\Delta}_t| \leq \frac{1}{m}$.

Note that $f_{t+1}(x) = \tilde{f}_{t+1}(x)$ for $x \notin B_t$ and $f_{t+1}(x) = \tilde{f}_{t+1}(x) + \Delta_t - \tilde{\Delta}_t$ for $x \in B_t$ where $|\Delta_t - \tilde{\Delta}_t| \leq \frac{1}{m}$. Applying Lemma 3.1 with $\Delta = \Delta_t - \tilde{\Delta}_t$, $\alpha \leq \frac{\rho}{m}$, and $(f, f') = (f_{t+1}, \tilde{f}_{t+1})$, we have that

$$PB_q^{\mathcal{S}}(f_{t+1}) - PB_q^{\mathcal{S}}(\tilde{f}_{t+1}) = \Pr_{(x,s)\sim\mathcal{S}}[x \in B_t] \cdot \left(PB^{\mathcal{S}|B_t}(f_{t+1}) - PB^{\mathcal{S}|B_t}(\tilde{f}_{t+1})\right)$$

$$\leq \Pr_{(x,s)\sim\mathcal{S}}[x \in B_t] \cdot \left(\frac{\rho}{m}\frac{1}{m} - \left(\frac{\rho}{m}\right)^2\frac{1}{2\rho}\right)$$

$$\leq \frac{\rho}{m^2}.$$

**(4) Combining (\*) and (\*\*):** We get

$$PB_q^{\mathcal{S}}(f_{t+1}) - PB_q^{\mathcal{S}}(f_t) \leq -\frac{\alpha}{4\rho m} + \frac{\rho}{m^2}$$

$$= -\frac{\alpha^2}{32\rho^3}.$$

**(5) Guarantees:** It directly follows that

$$PB_q^{\mathcal{S}}(f_T) - PB_q^{\mathcal{S}}(f_0) \leq T\frac{\alpha^2}{32\rho^3}.$$

With $0 \leq PB_q^{\mathcal{S}}(f) \leq 1$ for any $f : \mathcal{X} \to [0,1]$, we note that $f_T$ must be $\alpha$-approximately quantile multicalibrated with respect to $\mathcal{G}$ and $q$ after at most $\frac{32\rho^3}{\alpha^2}$ many rounds. $\qquad\square$

# C  Missing Details from Section 4

## C.1  On the i.i.d. versus exchangeability assumption

**Remark C.1.** *In this section, we prove PAC-like generalization theorems for our algorithms under the assumption that the data is drawn i.i.d. from some underlying distribution. Exchangeability is a weaker condition than independence, and requires only that the probability of observing any sequence of data is permutation invariant. De Finetti's representation theorem (Ressel, 1985) states that any infinite exchangeable sequence of data can be represented as a mixture of constituent distributions, each of which is i.i.d. Thus, our generalization theorems carry over from the i.i.d. setting to the more general exchangeable data setting: we can simply apply our theorems to each (i.i.d.) mixture component in the De Finetti representation of the exchangeable distribution. If for every mixture component, the model we learn has low quantile multicalibration error with probability $1 - \delta$ when the data is drawn from that component, then our models with also have low quantile multicalibration error with probability $1-\delta$ in expectation over the choice of the mixture component.*

## C.2  Helpful Concentration Bounds

**Theorem C.1** (Additive Chernoff Bound). *Let $\{X_i\}_{i=1}^n$ be independent random variables bounded such that for each $i \in [n]$, $X_i \in [0,1]$. Let $S_n = \sum_{i=1}^n X_i$ denote their sum. Then for all $\epsilon > 0$,*

$$\Pr_{\{X_i\}_{i=1}^n}[|S_n - \mathbb{E}[S_n]| \geq \epsilon] \leq 2\exp\left(-\frac{\epsilon^2}{n}\right).$$

**Theorem C.2** (Multiplicative Chernoff Bound). *Let $\{X_i\}_{i=1}^n$ be independent random variables bounded such that for each $i \in [n]$, $X_i \in [0,1]$. Let $S_n = \sum_{i=1}^n X_i$ denote their sum. Then for all $\eta > 0$,*

$$\Pr_{\{X_i\}_{i=1}^n}[|S_n - \mathbb{E}[S_n]| \geq \eta\,\mathbb{E}[S_n]] \leq 2\exp\left(-\frac{\mathbb{E}[S_n]\eta^2}{3}\right).$$

**Lemma C.1.** *Fix any $B \subseteq \mathcal{X}$. Given $S = \{(x_i, s_i)\}_{i=1}^n \sim \mathcal{S}^n$, we have*

$$\left|\frac{1}{n}\sum_{i=1}^n \mathbb{1}[x_i \in B] - \Pr_{(x,s)}[x \in B]\right| \leq \sqrt{\frac{3\ln(\frac{2}{\delta})\Pr_{\mathcal{S}}[x \in B]}{n}}.$$

*Proof.* This is just a direct application of the Chernoff bound (Theorem C.2) where we set $\eta = \sqrt{\frac{3\ln(\frac{2}{\delta})}{n\Pr_{\mathcal{S}}[x \in B]}}$. $\qquad\square$

**Lemma C.2.** *Fix any $B \subseteq \mathcal{X}$ and $f : \mathcal{X} \to [0,1]$. Given $S = \{(x_i, s_i)\}_{i=1}^n \sim \mathcal{S}^n$, we have*

$$\left|\frac{1}{n}\sum_{i=1}^n \mathbb{1}[s \leq f(x), x_i \in B] - \Pr_{(x,s)}[s \leq f(x), x \in B]\right| \leq \sqrt{\frac{3\ln(\frac{2}{\delta})\Pr_{\mathcal{S}}[s \leq f(x), x \in B]}{n}} \leq \sqrt{\frac{3\ln(\frac{2}{\delta})\Pr_{\mathcal{S}}[x \in B]}{n}}.$$

*Proof.* This is just a direct application of the Chernoff bound (Theorem C.2) where we set $\eta = \sqrt{\frac{3\ln(\frac{2}{\delta})}{n\Pr_{\mathcal{S}}[s\leq f(x), x\in B]}}$. $\square$

**Lemma C.3.** *Fix any $B \subseteq \mathcal{X}$ and $f : \mathcal{X} \to [0,1]$. Suppose $\sqrt{\frac{3\ln(\frac{4}{\delta})}{n\Pr[x\in B]}} \leq \frac{1}{2}$. Given $S \sim \mathcal{D}^n$, we have that with probability $1-\delta$*

$$\left|\Pr_{(x,s)\sim\tilde{\mathcal{S}}|B}[s \leq f(x)] - \Pr_{(x,s)\sim\mathcal{S}|B}[s \leq f(x)]\right| \leq 5\sqrt{\frac{3\ln(\frac{4}{\delta})}{n\Pr_{\mathcal{S}}[x\in B]}}$$

*Proof.* Note that

$$\Pr_{(x,s)\sim\tilde{\mathcal{S}}|B}[s \leq f(x)] = \frac{\frac{1}{n}\sum_{i=1}^n \mathbb{1}[s_i \leq f(x_i), x_i \in B]}{\frac{1}{n}\sum_{i=1}^n \mathbb{1}[x_i \in B]}.$$

Lemma C.1 and C.2 together give us that with probability $1-\delta$,

$$\left|\frac{1}{n}\sum_{i=1}^n \mathbb{1}[s_i \leq f(x_i), x_i \in B] - \Pr_{(x,s)}[s \leq f(x), x \in B]\right| \leq \sqrt{\frac{3\ln(\frac{4}{\delta})\Pr_{\mathcal{S}}[s\leq f(x), x\in B]}{n}} \leq \sqrt{\frac{3\ln(\frac{4}{\delta})\Pr_{\mathcal{S}}[x\in B]}{n}}$$

$$\left|\frac{1}{n}\sum_{i=1}^n \mathbb{1}[x_i \in B] - \Pr_{(x,s)}[x \in B]\right| \leq \sqrt{\frac{3\ln(\frac{4}{\delta})\Pr_{\mathcal{S}}[x\in B]}{n}}.$$

With $\epsilon = \sqrt{\frac{3\ln(\frac{4}{\delta})\Pr_{\mathcal{S}}[x\in B]}{n}}$, we can show that

$$\Pr_{(x,s)\sim\tilde{\mathcal{S}}|B}[s \leq f(x)]$$
$$= \frac{\frac{1}{n}\sum_{i=1}^n \mathbb{1}[s_i \leq f(x_i), x_i \in B]}{\frac{1}{n}\sum_{i=1}^n \mathbb{1}[x_i \in B]}$$
$$\leq \frac{\Pr_{\mathcal{S}}[s \leq f(x), x \in B] + \epsilon}{\Pr_{\mathcal{S}}[x \in B] - \epsilon}$$
$$= \frac{\Pr_{\mathcal{S}}[s \leq f(x), x \in B] + \epsilon}{\Pr_{\mathcal{S}}[x \in B]\left(1 - \frac{\epsilon}{\Pr_{\mathcal{S}}[x\in B]}\right)}$$
$$\underbrace{\leq}_{(*)} \left(1 + \frac{2\epsilon}{\Pr_{\mathcal{S}}[x \in B]}\right)\left(\frac{\Pr_{\mathcal{S}}[s \leq f(x), x \in B] + \epsilon}{\Pr_{\mathcal{S}}[x \in B]}\right)$$
$$\leq \Pr_{\mathcal{S}|B}[s \leq f(x)] + \frac{\epsilon}{\Pr_{\mathcal{S}}[x \in B]} + \frac{2\epsilon}{\Pr_{\mathcal{S}}[x \in B]} + \frac{2\epsilon^2}{\Pr_{\mathcal{S}}[x \in B]^2}$$
$$\leq \Pr_{\mathcal{S}|B}[s \leq f(x)] + 3\sqrt{\frac{3\ln(\frac{4}{\delta})}{n\Pr_{\mathcal{S}}[x \in B]}} + 2\frac{3\ln(\frac{4}{\delta})}{n\Pr_{\mathcal{S}}[x \in B]}$$
$$\underbrace{\leq}_{(**)} \Pr_{\mathcal{S}|B}[s \leq f(x)] + 5\sqrt{\frac{3\ln(\frac{4}{\delta})}{n\Pr_{\mathcal{S}}[x \in B]}}$$

where for (*), we rely on the assumption that $\sqrt{\frac{3\ln(\frac{4}{\delta})}{n\Pr[x\in B]}} \leq \frac{1}{2}$ to apply the inequality $1/(1-x) \leq (1+2x)$ for $0 \leq x \leq 1/2$ and for (**), we rely on $\frac{3\ln(\frac{4}{\delta})}{n\Pr_{\mathcal{S}}[x\in B]} \leq 1$ to get $\sqrt{\frac{3\ln(\frac{4}{\delta})}{n\Pr_{\mathcal{S}}[x\in B]}} \geq \frac{3\ln(\frac{4}{\delta})}{n\Pr_{\mathcal{S}}[x\in B]}$. $\square$

**Lemma C.4.** *Fix any $B \subseteq \mathcal{X}$ and $f : \mathcal{X} \to [0,1]$. Suppose $\sqrt{\frac{3\ln(\frac{4}{\delta})}{n\Pr[x\in B]}} \leq \frac{1}{2}$. Given $S \sim \mathcal{D}^n$, we have that with probability $1-\delta$*

$$\left|\left(q - \Pr_{(x,s)\sim\mathcal{S}|B}[s \leq f(x)]\right)^2 - \left(q - \Pr_{(x,s)\sim\tilde{\mathcal{S}}|B}[s \leq f(x)]\right)^2\right| \leq 20\sqrt{\frac{3\ln(\frac{4}{\delta})}{n\Pr_{\mathcal{S}}[x \in B]}}$$

*for any $q \in [0, 1]$.*

*Proof.*

$$
\left| \left( q - \Pr_{(x,s)\sim\mathcal{S}|B}[s \le f_t(x)] \right)^2 - \left( q - \Pr_{(x,s)\sim\tilde{\mathcal{S}}|B}[s \le f_t(x)] \right)^2 \right|
$$

$$
= \left| 2q \left( \Pr_{\tilde{\mathcal{S}}|B}[s \le f(x)] - \Pr_{\mathcal{S}|B}[s \le f(x)] \right) + \left( \Pr_{\mathcal{S}|B}[s \le f(x)]^2 - \Pr_{\tilde{\mathcal{S}}|B}[s \le f(x)]^2 \right) \right|
$$

$$
\le 2 \left| \Pr_{\mathcal{S}|B}[s \le f(x)] - \Pr_{\tilde{\mathcal{S}}|B}[s \le f(x)] \right| + \left| \left( \Pr_{\mathcal{S}|B}[s \le f(x)] - \Pr_{\tilde{\mathcal{S}}|B}[s \le f(x)] \right) \left( \Pr_{\mathcal{S}|B}[s \le f(x)] + \Pr_{\tilde{\mathcal{S}}|B}[s \le f(x)] \right) \right|
$$

$$
\le 4 \left| \Pr_{\mathcal{S}|B}[s \le f(x)] - \Pr_{\tilde{\mathcal{S}}|B}[s \le f(x)] \right|
$$

$$
\le 20 \sqrt{\frac{3\ln(\frac{4}{\delta})}{n\Pr_{\mathcal{S}}[x \in B]}}.
$$

where we use Lemma C.3 for the last inequality. $\qquad\square$

**Lemma C.5.** *Fix $B \subseteq \mathcal{X}, v \in [\frac{1}{m}], g \in \mathcal{G}$, and $f : \mathcal{X} \to [0,1]$. Given $S \sim \mathcal{D}^n$, we have with probability $1 - \delta$*

$$
\left| \Pr_{(x,s)\sim\mathcal{S}}[x \in B] \left( q - \Pr_{(x,s)\sim\mathcal{S}|B}[s \le f(x)] \right)^2 - \Pr_{(x,s)\sim\tilde{\mathcal{S}}}[x \in B] \left( q - \Pr_{(x,s)\sim\tilde{\mathcal{S}}|B}[s \le f(x)] \right)^2 \right|
$$

$$
\le 21 \sqrt{\frac{3\ln(\frac{8}{\delta})\Pr_{\mathcal{S}}[x \in B]}{n}} + \frac{12\ln(\frac{8}{\delta})}{n}.
$$

*Proof.* First, Suppose $\sqrt{\frac{3\ln(\frac{8}{\delta})}{n\Pr_{\mathcal{S}}[x\in B]}} > \frac{1}{2}$. Then, with probability $1 - \delta$,

$$
\Pr_{(x,s)\sim\tilde{\mathcal{S}}}[B] \left( q - \Pr_{(x,s)\sim\tilde{\mathcal{S}}|B}[s \le f(x)] \right)^2
$$

$$
\le \left( \Pr_{(x,s)\sim\mathcal{S}}[B] + \sqrt{\frac{3\ln(\frac{2}{\delta})\Pr_{\mathcal{S}}[x \in B]}{n}} \right) \left( q - \Pr_{(x,s)\sim\tilde{\mathcal{S}}|B}[s \le f(x)] \right)^2 \qquad\qquad \text{Lemma C.1}
$$

$$
\le \Pr_{(x,s)\sim\mathcal{S}}[B] + \sqrt{\frac{3\ln(\frac{8}{\delta})\Pr_{\mathcal{S}}[x \in B]}{n}} \qquad\qquad\qquad \left( q - \Pr_{(x,s)\sim\tilde{\mathcal{S}}|B}[s \le f(x)] \right)^2 \le 1
$$

$$
\le \frac{12\ln(\frac{8}{\delta})}{n} + \sqrt{\frac{3\ln(\frac{8}{\delta})\Pr_{\mathcal{S}}[x \in B]}{n}} \qquad\qquad\qquad\qquad \sqrt{\frac{3\ln(\frac{8}{\delta})}{n\Pr_{\mathcal{S}}[x \in B]}} > \frac{1}{2}.
$$

On the other hand, we have

$$
\Pr_{(x,s)\sim\mathcal{S}}[B] \left( q - \Pr_{(x,s)\sim\mathcal{S}|B}[s \le f(x)] \right)^2 \le \Pr_{(x,s)\sim\mathcal{S}}[B] \le \frac{12\ln(\frac{8}{\delta})}{n}.
$$

Because $|a - b| \le \max(a, b)$ for $a, b \in [0, 1]$, we have

$$
\left| \Pr_{(x,s)\sim\mathcal{S}}[B] \left( q - \Pr_{(x,s)\sim\mathcal{S}|B}[s \le f(x)] \right)^2 - \Pr_{(x,s)\sim\tilde{\mathcal{S}}}[B] \left( q - \Pr_{(x,s)\sim\tilde{\mathcal{S}}|B}[s \le f(x)] \right)^2 \right|
$$

$$\leq \sqrt{\frac{3\ln(\frac{8}{\delta})\Pr_{\mathcal{S}}[x \in B]}{n}} + \frac{12\ln(\frac{8}{\delta})}{n}.$$

Now, suppose otherwise: $\sqrt{\frac{3\ln(\frac{8}{\delta})}{n\Pr[x \in B]}} \leq \frac{1}{2}$. Then, Lemma C.1 and C.4 promise us that with probability $1 - \delta$,

$$\Pr_{(x,s)\sim\tilde{\mathcal{S}}}[B]\left(q - \Pr_{(x,s)\sim\tilde{\mathcal{S}}|B}[s \leq f(x)]\right)^2$$

$$\leq \left(\Pr_{(x,s)\sim\mathcal{S}}[B] + \epsilon_1\right)\left(\left(q - \Pr_{(x,s)\sim\mathcal{S}|B}[s \leq f(x)]\right)^2 + \epsilon_2\right)$$

$$\leq \Pr_{(x,s)\sim\mathcal{S}}[B]\left(q - \Pr_{(x,s)\sim\mathcal{S}|B}[s \leq f(x)]\right)^2 + \Pr_{(x,s)\sim\mathcal{S}}[B]\epsilon_2 + \epsilon_1 + \epsilon_1\epsilon_2$$

where $\epsilon_1 = \sqrt{\frac{3\ln(\frac{4}{\delta})\Pr_{\mathcal{S}}[x \in B]}{n}}$. and $\epsilon_2 = 20\sqrt{\frac{3\ln(\frac{8}{\delta})}{n\Pr_{\mathcal{S}}[x \in B]}}$.

We can show that

$$\Pr_{(x,s)\sim\mathcal{S}}[B]\epsilon_2 + \epsilon_1 + \epsilon_1\epsilon_2$$

$$\leq 21\sqrt{\frac{3\ln(\frac{8}{\delta})\Pr_{\mathcal{S}}[x \in B]}{n}} + \frac{3\ln(\frac{8}{\delta})}{n}.$$

The opposite direction works the same way. $\qquad\square$

## C.3 Out of sample guarantees for BatchGCP

**Theorem C.3.** *Suppose $\mathcal{S}$ is $\rho$-Lipschitz, and write $\hat{f}(\cdot; \lambda^*) = \texttt{BatchGCP}(f, \mathcal{G}, q, \mathcal{D})$ and $\hat{f}(\cdot; \lambda_D^*) = \texttt{BatchGCP}(f, \mathcal{G}, q, D)$ given some $D = \{(x_i, y_i)\}_{i=1}^n$ drawn from $\mathcal{D}$. Then we have with probability $1 - \delta$ over the randomness of drawing $D$ from $\mathcal{D}$*

$$\left|\Pr_{(x,y)\sim\mathcal{D}}[y \in \mathcal{T}^{\hat{f}(\cdot,\lambda_D^*)}(x)|g(x=1)] - q\right| \leq \sqrt{\frac{\alpha'}{\Pr[g(x) = 1]}}$$

*where $\mathcal{T}^{\hat{f}(\cdot,\lambda_D^*)}(x) = \{y : s(x, y) \leq \hat{f}(x; \lambda_D^*)\}$, $q' = \max(q, 1 - q)$, $b = \lceil\max(||\lambda^*||_2, ||\lambda_D^*||_2)\rceil$, and $\alpha' = 24\rho b q'\sqrt{\frac{\ln(\frac{\pi^2 b^2}{3\delta}) + |\mathcal{G}|\ln(1+2n)}{2n}}$.*

*Proof.* Fix some $B \in \mathbb{N}$. For any fixed $\lambda$ where $||\lambda||_2 \leq B$, the Chernoff Bound (Theorem C.1) with rescaling gives us that with probability $1 - \delta$ over the randomness of drawing $D = \{(x_i, y_i)\}_{i=1}^n$ from $\mathcal{D}$,

$$\left|\frac{1}{n}\sum_{i=1}^n L_q(\hat{f}(x_i; \lambda), y_i) - \mathbb{E}_{(x,y)\sim\mathcal{D}}[L_q(\hat{f}(x; \lambda), y)]\right| \leq 4q'B\sqrt{\frac{\ln(\frac{2}{\delta})}{2n}} \tag{1}$$

as $\lambda \to L_q(\hat{f}(x, \lambda), y)$ is $q'$-Lipschitz.

We now show how to union bound the above concentration over all $\lambda$ where $||\lambda||_2 \leq B$. To do so, we first create a finite $\epsilon$-net for a ball of radius $B$ and union-bound over each $\lambda$ in the net. Then we can argue that due to the Lipschitzness of the pinball loss $L_q$, the empirical pinball loss with respect to each $\lambda$ that is in the ball must be concentrated around its distributional loss.

We fist provide the standard $\epsilon$-net argument:

**Lemma C.6.** *Fix $\epsilon \in \mathbb{R}$ and $B \in \mathbb{N}$. There exists a $R_B = \{\lambda_1, \ldots, \lambda_{k_B}\}$ where $k_B \leq \left(1 + \frac{2B}{\epsilon}\right)^{|\mathcal{G}|}$ such that the following holds true: for every $\lambda$ where $||\lambda||_2 \leq B$, there exists $j \in [k]$ such that $||\lambda - \lambda_j||_2 \leq \epsilon$.*

*Proof.* For simplicity, write $P_B = \{\lambda \in \mathbb{R}^{|\mathcal{G}|} : ||\lambda||_2 \leq B\}$.

We say that a set $R \subseteq \mathbb{R}^{|\mathcal{G}|}$ is an $\epsilon$-net of $P_B$ if for every $\lambda \in P_B$, there exists $\lambda' \in R$ such that $||\lambda - \lambda'||_2 \leq \epsilon$.

Without loss of generality, we suppose $B = 1$. Since an $\frac{\epsilon}{B}$-cover of $P_1$ can be scaled up to an $\epsilon$-cover of $P_B$ — i.e. given $R_1$ which is an $\frac{\epsilon}{B}$-cover of $P_1$, the following set $R_B = \{B \cdot \lambda : \lambda \in R_1\}$ is an $\epsilon$-cover of $P_B$.

Now, choose a set $R$ to be a maximally $\epsilon$-separated subset of $P_1$: for every $u, v \in B$, $||u - v|| \geq \epsilon$ and no set $R'$ such that $R \subset R'$ has this property.

Due to its maximal property, $R$ must be an $\epsilon$-cover for $P_1$. Otherwise, it means that there exists some point $u \in P_1$ such that for every $v \in R$ $||u - v|| > \epsilon$. Note that $R \cup \{u\}$ would still be an $\epsilon$-separate subset of $P_1$, contradicting the maximality of $R$.

Due to the $\epsilon$-separability of $R$, we have that the balls centered at each $u \in R$ with radius of $\frac{\epsilon}{2}$ is all disjoint, meaning the sum of the volume of these balls is the volume of their union. On the other hand, we have that they all lie in a ball of radius $1 + \frac{\epsilon}{2}$, $P_{1+\frac{\epsilon}{2}}$. Therefore, we have

$$vol(P_{\frac{\epsilon}{2}}) \cdot |R| \leq vol(P_{1+\frac{\epsilon}{2}}).$$

Since $vol(P_c) = c^{|\mathcal{G}|} \cdot vol(P_1)$, we have that

$$|R| \leq \frac{(1 + \frac{\epsilon}{2})^{|\mathcal{G}|}}{(\frac{\epsilon}{2})^{|\mathcal{G}|}} = \left(1 + \frac{2}{\epsilon}\right)^{|\mathcal{G}|}.$$

$\square$

By union-bounding inequality (1) over $R_B$, we have that with probability $1 - \delta$

$$\max_{j \in [k_B]} \left| \frac{1}{n} \sum_{i=1}^{n} L_q(\hat{f}(x_i; \lambda_j), y_i) - \mathbb{E}_{(x,y)\sim\mathcal{D}}[L_q(\hat{f}(x; \lambda_j), y)] \right| \leq 4q'B\sqrt{\frac{\ln(\frac{2k_B}{\delta})}{2n}}.$$

Using $q'$-Lipschitzness of $L_q$, we can further show that for any $\lambda$ where $||\lambda||_2 \leq B$, we have

$$\left| \frac{1}{n} \sum_{i=1}^{n} L_q(\hat{f}(x_i; \lambda), y_i) - \mathbb{E}_{(x,y)\sim\mathcal{D}}[L_q(\hat{f}(x; \lambda), y)] \right|$$

$$= \left| \frac{1}{n} \sum_{i=1}^{n} L_q(\hat{f}(x_i; \lambda), y_i) - \frac{1}{n} \sum_{i=1}^{n} L_q(\hat{f}(x_i; \lambda_j), y_i) + \frac{1}{n} \sum_{i=1}^{n} L_q(\hat{f}(x_i; \lambda_j), y_i)] \right.$$

$$\left. + \mathbb{E}_{(x,y)\sim\mathcal{D}}[L_q(\hat{f}(x; \lambda_j), y)] - \mathbb{E}_{(x,y)\sim\mathcal{D}}[L_q(\hat{f}(x; \lambda_j), y)] - \mathbb{E}_{(x,y)\sim\mathcal{D}}[L_q(\hat{f}(x; \lambda), y)] \right|$$

$$\leq \left| \frac{1}{n} \sum_{i=1}^{n} L_q(\hat{f}(x_i; \lambda), y_i) - \frac{1}{n} \sum_{i=1}^{n} L_q(\hat{f}(x_i; \lambda_j), y_i)] \right|$$

$$+ \left| \frac{1}{n} \sum_{i=1}^{n} L_q(\hat{f}(x_i; \lambda_j), y_i)) - \mathbb{E}_{(x,y)\sim\mathcal{D}}[L_q(\hat{f}(x; \lambda_j), y] \right|$$

$$+ \left| \mathbb{E}_{(x,y)\sim\mathcal{D}}[L_q(\hat{f}(x; \lambda_j), y)]) - \mathbb{E}_{(x,y)\sim\mathcal{D}}[L_q(\hat{f}(x; \lambda), y)] \right|$$

$$\leq 2\epsilon q' + 4q'B\sqrt{\frac{\ln(\frac{2}{\delta}) + \ln(k_B)}{2n}}$$

$$= 2\epsilon q' + 4q'B\sqrt{\frac{\ln(\frac{2}{\delta}) + |\mathcal{G}|\ln(1 + \frac{2B}{\epsilon})}{2n}}$$

where $j$ is chosen such that $||\lambda - \lambda_j||_2 \leq \epsilon$ and we can find such $j$ because $R_B$ is an $\epsilon$-net for the ball of radius $B$.

Setting $\epsilon = \frac{B}{n}$ yields

$$\leq \frac{2Bq'}{n} + 4q'B\sqrt{\frac{\ln(\frac{2}{\delta}) + |\mathcal{G}|\ln(1+2n)}{2n}} \leq 6Bq'\sqrt{\frac{\ln(\frac{2}{\delta}) + |\mathcal{G}|\ln(1+2n)}{2n}}$$

for sufficiently large $n$.

We set $\delta_b = \frac{6\delta}{\pi^2 b^2}$ so that $\sum_{b=1}^{\infty} \delta_b = \delta$. In other words, with probability $1 - \delta$, we have simultaneously over all $b \in \mathbb{N}$ and $\lambda$ where $||\lambda||_2 \leq b$

$$\left| \frac{1}{n}\sum_{i=1}^{n} L_q(\hat{f}(x_i; \lambda), y_i) - \mathop{\mathbb{E}}_{(x,y)\sim\mathcal{D}}[L_q(\hat{f}(x; \lambda), y)] \right| \leq 6bq'\sqrt{\frac{\ln(\frac{\pi^2 b^2}{3\delta}) + |\mathcal{G}|\ln(1+2n)}{2n}}.$$

In other words, the final output $\lambda_D^*$ from `BatchGCP` and $\lambda^*$ which is the optimal solution with respect to the true distribution $\mathcal{D}$ must be such that

$$\left| \frac{1}{n}\sum_{i=1}^{n} L_q(\hat{f}(x_i; \lambda_D^*), y_i) - \mathop{\mathbb{E}}_{(x,y)\sim\mathcal{D}}[L_q(\hat{f}(x; \lambda_D^*), y)] \right| \leq 6bq'\sqrt{\frac{\ln(\frac{\pi^2 b^2}{3\delta}) + |\mathcal{G}|\ln(1+2n)}{2n}}$$

$$\left| \frac{1}{n}\sum_{i=1}^{n} L_q(\hat{f}(x_i; \lambda^*), y_i) - \mathop{\mathbb{E}}_{(x,y)\sim\mathcal{D}}[L_q(\hat{f}(x; \lambda^*), y)] \right| \leq 6bq'\sqrt{\frac{\ln(\frac{\pi^2 b^2}{3\delta}) + |\mathcal{G}|\ln(1+2n)}{2n}}.$$

where $b = \lceil \max(||\lambda^*||_2, ||\lambda_D^*||_2) \rceil$. In other words,

$$\mathop{\mathbb{E}}_{(x,y)\sim\mathcal{D}}[L_q(\hat{f}(x; \lambda_D^*), y)] - \mathop{\mathbb{E}}_{(x,y)\sim\mathcal{D}}[L_q(\hat{f}(x; \lambda^*), y)] \leq 12bq'\sqrt{\frac{\ln(\frac{\pi^2 b^2}{3\delta}) + |\mathcal{G}|\ln(1+2n)}{2n}}.$$

Now, for the sake of contradiction, suppose that there exists some $g \in \mathcal{G}$ such that

$$\mathop{\Pr}_{(x,s)\sim\mathcal{S}}[g(x) = 1] \cdot \left( q - \mathop{\Pr}_{(x,s)\sim\mathcal{S}}[s \leq \hat{f}(x; \lambda_D^*) | g(x) = 1] \right)^2 > \alpha'$$

where $\alpha' = 24\rho bq'\sqrt{\frac{\ln(\frac{\pi^2 b^2}{3\delta}) + |\mathcal{G}|\ln(1+2n)}{2n}}$.

Then, Lemma 3.2 tells us that we can decrease the true pinball loss with respect to $\lambda_D^*$ by at least $\frac{\alpha'}{2\rho}$ by patching $B = \{x : g(x) = 1\}$. However, that cannot be the case that as that would mean there exists $\lambda'$ such that

$$\mathop{\mathbb{E}}_{(x,y)\sim\mathcal{D}}[L_q(\hat{f}(x; \lambda'), y)] < \mathop{\mathbb{E}}_{(x,y)\sim\mathcal{D}}[L_q(\hat{f}(x; \lambda^*), y)]$$

where $\lambda'$ is such that $\lambda'_g = \lambda_{D,g}^*$ for all $g' \neq g$ and $\lambda'_g$ is chosen to satisfy

$$q = \mathop{\Pr}_{(x,y)\sim\mathcal{D}}[f(x) + \lambda'_g | g(x) = 1].$$

This is a contradiction as we have already defined

$$\lambda^* = \arg\min_\lambda \mathop{\mathbb{E}}_{(x,y)\sim\mathcal{D}}[L_q(\hat{f}(x; \lambda), y)].$$

$\square$

## C.4 OUT OF SAMPLE GUARANTEES FOR BATCHMVP

### C.4.1 OUT-OF-SAMPLE QUANTILE MULTICALIBRATION BOUND FOR FIXED $T$

**Theorem 4.1.** *Suppose $\mathcal{S}$ is $\rho$-Lipschitz and $S \sim \mathcal{S}^n$. Suppose `BatchMVP`$(f, \alpha, q, \mathcal{G}, \rho, \tilde{\mathcal{S}}_S, m)$ (Algorithm 2) runs for $T$ rounds and outputs model $f_T$. Then $f_T$ is $\alpha'$-approximately $q$-quantile multicalibrated with respect to $\mathcal{G}$ on $\mathcal{S}$ with probability $1 - \delta$, where*

$$\alpha' = \alpha + 21\sqrt{\frac{3\rho^2\left(\ln(\frac{4\pi^2 T^2}{3\delta}) + T\ln(\frac{\rho^4|\mathcal{G}|}{\alpha^2})\right)}{2\alpha n}} + \frac{12\rho^2(\frac{4\pi^2 T^2}{3\delta}) + T\ln(\frac{\rho^4|\mathcal{G}|}{\alpha^2}))}{\alpha n}.$$

*Proof.* For each $t \in \mathbb{N}$, define $\delta_t = \delta \cdot \frac{6}{\pi^2} \cdot \frac{1}{t^2}$. Note that

$$\sum_{t=1}^{\infty} \delta_t = \delta \frac{6}{\pi^2} \sum_{t=1}^{\infty} \frac{1}{t^2} = \delta$$

as $\sum_{t=1}^{\infty} \frac{1}{t^2} = \frac{\pi^2}{6}$.

Fixed any $t \in \mathbb{N}$ and $\delta$. Union-bounding Lemma C.5 over $g \in \mathcal{G}$ and $f \in \mathcal{C}_t$, we have that for any $g \in \mathcal{G}$, with probability $1 - \delta$,

$$\sum_{v \in R(f_t)} \Pr_{(x,s) \sim \mathcal{S}}[f_t(x) = v, g(x) = 1] \left( q - \Pr_{(x,s) \sim \mathcal{S}}[s \leq f_t(x) | f_t(x) = v, g(x) = 1] \right)^2$$

$$\leq \sum_{v \in R(f_t)} \Pr_{(x,s) \sim \tilde{\mathcal{S}}}[f_t(x) = v, g(x) = 1] \left( q - \Pr_{(x,s) \sim \tilde{\mathcal{S}}}[s \leq f_t(x) | f_t(x) = v, g(x) = 1] \right)^2$$

$$+ \underbrace{\sum_{v \in R(f_t)} 21 \sqrt{\frac{3 \left( \ln(\frac{8}{\delta}) + t \ln(4m^2 |\mathcal{G}|) \right) \Pr_{\mathcal{S}}[f_t(x) = v, g(x) = 1]}{n}} + \frac{12(\ln(\frac{8}{\delta}) + t \ln(4m^2 |\mathcal{G}|))}{n}}_{(*)}.$$

We can further bound $(*)$ as

$$(*) \leq \sum_{v \in R(f_t)} 21 \sqrt{\frac{3 \left( \ln(\frac{8}{\delta}) + t \ln(4m^2 |\mathcal{G}|) \right) \Pr_{\mathcal{S}}[f_t(x) = v, g(x) = 1]}{n}} + 2m \frac{12(\ln(\frac{8}{\delta}) + t \ln(4m^2 |\mathcal{G}|))}{n}$$

$$\underbrace{\leq}_{(**)} \alpha + 21 \sqrt{\frac{3 \left( \ln(\frac{8}{\delta}) + t \ln(4m^2 |\mathcal{G}|) \right) \frac{\Pr_{\mathcal{S}}[g(x)=1]}{|R(f_t)|}}{n} |R(f)|^2} + \frac{12\rho^2(\ln(\frac{8}{\delta}) + t \ln(4m^2 |\mathcal{G}|))}{\alpha n}$$

$$\leq 21 \sqrt{\frac{3 \left( \ln(\frac{8}{\delta}) + t \ln(4m^2 |\mathcal{G}|) \right) |R(f_t)|}{n}} + \frac{12\rho^2(\ln(\frac{8}{\delta}) + T \ln(4m^2 |\mathcal{G}|))}{\alpha n}$$

$$= 21 \sqrt{\frac{3\rho^2 \left( \ln(\frac{8}{\delta}) + t \ln(4m^2 |\mathcal{G}|) \right)}{4\alpha n}} + \frac{12\rho^2(\ln(\frac{8}{\delta}) + t \ln(4m^2 |\mathcal{G}|))}{\alpha n}$$

$$= 21 \sqrt{\frac{3\rho^2 \left( \ln(\frac{8}{\delta}) + t \ln(\frac{\rho^4 |\mathcal{G}|}{\alpha^2}) \right)}{2\alpha n}} + \frac{12\rho^2(\ln(\frac{8}{\delta}) + t \ln(\frac{\rho^4 |\mathcal{G}|}{\alpha^2}))}{\alpha n}$$

where we used $|R(f_t)| = m + 1 \leq 2m$ and $m = \frac{\rho^2}{2\alpha}$. Inequality $(**)$ follows from the fact that $h(z) = \sqrt{\frac{3t \left( \ln(\frac{8}{\delta}) + t \ln(4m^2 |\mathcal{G}|) \right) \cdot z}{n}}$ is concave and so the optimization problem $\sum_{i=1}^{|R(f_t)|} h(z_i)$ where $\sum_{i=1}^{|R(f_t)|} z_i = \Pr_{\mathcal{S}}(g(x) = 1)$ is maximized at $z_i = \frac{\Pr_{\mathcal{S}}(g(x)=1)}{|R(f_t)|}$ for each $i \in [|R(f_t)|]$.

In other words, with probability $1 - \delta = 1 - \sum_{t=1}^{\infty} \delta_t$, we simultaneously have for every $t \in \mathbb{N}$

$$\sum_{v \in R(f_t)} \Pr_{(x,s) \sim \mathcal{S}}[f_t(x) = v, g(x) = 1] \left( q - \Pr_{(x,s) \sim \mathcal{S}}[s \leq f_t(x) | f_t(x) = v, g(x) = 1] \right)^2$$

$$\leq \sum_{v \in R(f_t)} \Pr_{(x,s) \sim \tilde{\mathcal{S}}}[f_t(x) = v, g(x) = 1] \left( q - \Pr_{(x,s) \sim \tilde{\mathcal{S}}}[s \leq f_t(x) | f_t(x) = v, g(x) = 1] \right)^2$$

$$+ 21 \sqrt{\frac{3\rho^2 \left( \ln(\frac{8}{\delta_t}) + t \ln(\frac{\rho^4 |\mathcal{G}|}{\alpha^2}) \right)}{2\alpha n}} + \frac{12\rho^2(\ln(\frac{8}{\delta_t}) + t \ln(\frac{\rho^4 |\mathcal{G}|}{\alpha^2}))}{\alpha n}$$

$$\leq \sum_{v \in R(f_t)} \Pr_{(x,s) \sim \tilde{\mathcal{S}}}[f_t(x) = v, g(x) = 1] \left( q - \Pr_{(x,s) \sim \tilde{\mathcal{S}}}[s \leq f_t(x)|f_t(x) = v, g(x) = 1] \right)^2$$

$$+ 21\sqrt{\frac{3\rho^2 \left( \ln(\frac{4\pi^2 t^2}{3\delta}) + t\ln(\frac{\rho^4 |\mathcal{G}|}{\alpha^2}) \right)}{2\alpha n} + \frac{12\rho^2(\frac{4\pi^2 t^2}{3\delta}) + t\ln(\frac{\rho^4 |\mathcal{G}|}{\alpha^2}))}{\alpha n}}.$$

Finally, when our algorithm halts at round $T$, $f_T$ is $\alpha$-approximately multicalibrated with respect to its empirical distribution $\tilde{\mathcal{S}}$. Therefore, we have

$$\sum_{v \in R(f_T)} \Pr_{(x,s) \sim \mathcal{S}}[f_T(x) = v, g(x) = 1] \left( q - \Pr_{(x,s) \sim \mathcal{S}}[s \leq f_T(x)|f_T(x) = v, g(x) = 1] \right)^2$$

$$\leq \alpha + 21\sqrt{\frac{3\rho^2 \left( \ln(\frac{4\pi^2 t^2}{3\delta}) + t\ln(\frac{\rho^4 |\mathcal{G}|}{\alpha^2}) \right)}{2\alpha n} + \frac{12\rho^2(\frac{4\pi^2 t^2}{3\delta}) + t\ln(\frac{\rho^4 |\mathcal{G}|}{\alpha^2}))}{\alpha n}}.$$

$\square$

### C.4.2 Sample complexity of maintaining convergence speed of BatchMVP

We write $\mathcal{C}_0 = \{f_0\}$ and

$$\mathcal{C}_{t+1} = \left\{ f_{t+1} : \begin{array}{l} f_{t+1}(x) = \mathrm{Patch}(f_t, B(v, g), \Delta) \text{ where } B(v, g) = \{x : f_t(x) = v, g(x) = 1\} \\ \text{for all } f_t \in \mathcal{C}_t, v \in \left[\frac{1}{m}\right], g \in \mathcal{G}, \Delta \in \left[\frac{1}{m}\right]. \end{array} \right\}$$

to denote the set of all possible models $f_t$ we could obtain at round $t$ of Algorithm 2 regardless of what dataset $S = \{(x_i, s_i)\}_{i=1}^n$ is used as input.

**Lemma C.7.** *Fixing the initial model $f_0$, the number of distinct models $f_t$ that can arise at round $t$ of Algorithm 2 (quantified over all possible input datasets $S$) is upper bounded by:*

$$|\mathcal{C}_t| = ((m+1)^2 |\mathcal{G}|)^t \leq (4m^2 |\mathcal{G}|)^t.$$

*Proof.* The proof is by induction on $t \in \mathbb{N}$. By construction we have $|\mathcal{C}_0| = 1$. Now, note that $|\mathcal{C}_{t+1}| = (m+1)^2 |\mathcal{G}||\mathcal{C}_t|$ because we consider all possible combinations of $f_t \in \mathcal{C}_t, v \in \left[\frac{1}{m}\right], g \in \mathcal{G}$, and $\Delta \in \left[\frac{1}{m}\right]$. Therefore, if $|\mathcal{C}_t| = ((m+1)^2 |\mathcal{G}|)^t$ for some $t$, then $|\mathcal{C}_{t+1}| = ((m+1)^2 |\mathcal{G}|)^{t+1}$. $\square$

**Theorem 4.2.** *Suppose $\mathcal{S}$ is $\rho$-Lipschitz and continuous, $m = \frac{\rho^2}{2\alpha}$, and our calibration set $S \sim \mathcal{S}^n$ consists of $n$ iid. samples drawn from $\mathcal{S}$, where $n$ is sufficiently large: $n \geq 92928 \left( \ln \left( \frac{128\rho^3}{\alpha^2 \delta} \right) + \frac{8\rho^3}{\alpha^2} \ln \left( \frac{\rho^4 |\mathcal{G}|}{\alpha^2} \right) \right) \max \left( \frac{\rho^4}{4\alpha^4}, \frac{\rho^6}{\alpha^4} \right)$. Then $BatchMVP(f, \alpha, q, \mathcal{G}, \rho, \tilde{\mathcal{S}}_S, m)$ (Algorithm 2) halts after $T \leq \frac{8\rho^3}{\alpha^2}$ rounds with prob. $1 - \delta$.*

*Proof.* Fix any round $t$. Because $f_t$ is not $\alpha$-approximately quantile multicalibrated with respect to $\mathcal{G}$ and $q$ on $\tilde{\mathcal{S}}$, we have

$$\Pr_{(x,s) \sim \tilde{\mathcal{S}}}[x \in B_t] \left( q - \Pr_{(x,s) \sim \tilde{\mathcal{S}}}[s \leq f_t(x)|x \in B_t]) \right)^2 \geq \frac{\alpha}{m+1} \geq \frac{\alpha}{2m}.$$

Now, the patch operation in this can be decomposed into the following:

$$v_t \rightarrow v_t + \Delta^* \quad \text{where } q = \Pr_{(x,s) \sim \mathcal{S}|B_t}[s \leq f_t(x) + \Delta^*]$$

$$v_t + \Delta^* \rightarrow v_t + \tilde{\Delta}^* \quad \text{where } \tilde{\Delta}^* = \arg\min_{v \in [1/m]} |v - \Delta^*|$$

$$v_t + \tilde{\Delta}^* \rightarrow v_t + \Delta_t.$$

For convenience, we write

$$f_t^*(x) = f_t(x) + \Delta^* \cdot \mathbb{1}[x \in B_t]$$
$$\tilde{f}_t^*(x) = f_t(x) + \tilde{\Delta}^* \cdot \mathbb{1}[x \in B_t]$$
$$f_{t+1}(x) = f_t(x) + \Delta_t \cdot \mathbb{1}[x \in B_t]$$

Now, we show that we decrease the empirical pinball loss $PB^{\tilde{\mathcal{S}}}$ as we go from $f_t$ to $f_{t+1}$ in each round $t$ with high probability.

**(1) $f_t \to f_t^*$:** First, we can show progress in terms of the pinball loss on $\mathcal{S}$ and then by a Chernoff bound, we can show that we have made significant progress on $\tilde{\mathcal{S}}$ with high probability. More specifically, we must have decreased the pinball loss with respect to $\mathcal{S}|B_t$ by going from $v_t$ to $v_t + \Delta^*$. Note that $\Delta^*$ is chosen to satisfy the target quantile $q$ with respect to $\mathcal{S}|B_t$ and $\mathcal{S}$ is $\rho$-Lipschitz.

Because the empirical quantile error was significant for $(v_t, g_t)$, we can show that the true quantile error must have been significant. By union bounding Lemma C.5 over all $f \in \mathcal{C}_t$ and using Lemma C.7 to bound the cardinality of $\mathcal{C}_t$, we have with probability $1 - \delta$

$$\left| \Pr_{(x,s)\sim\mathcal{S}}[B_t] \left( q - \Pr_{(x,s)\sim\mathcal{S}|B_t}[s \leq f_t(x)] \right)^2 - \Pr_{(x,s)\sim\tilde{\mathcal{S}}}[B_t] \left( q - \Pr_{(x,s)\sim\tilde{\mathcal{S}}|B_t}[s \leq f_t(x)] \right)^2 \right|$$

$$\leq 21 \sqrt{\frac{3 \ln(\frac{8|\mathcal{C}_t|}{\delta}) \Pr_{\mathcal{S}}[x \in B]}{n}} + \frac{12 \ln(\frac{8|\mathcal{C}_t|}{\delta})}{n}$$

$$\leq 21 \sqrt{\frac{3 \left( \ln(\frac{8}{\delta}) + t \ln(4m^2 |\mathcal{G}|) \right) \Pr_{\mathcal{S}}[x \in B]}{n}} + \frac{12 \left( \ln(\frac{8}{\delta}) + t \ln(4m^2 |\mathcal{G}|) \right)}{n}$$

$$\leq 22 \sqrt{\frac{12 \left( \ln(\frac{8}{\delta}) + t \ln(4m^2 |\mathcal{G}|) \right)}{n}}$$

as long $n \geq 12 \left( \ln(\frac{8}{\delta}) + t \ln(4m^2 |\mathcal{G}|) \right)$. In other words, we have

$$\Pr_{(x,s)\sim\mathcal{S}}[B_t] \left( q - \Pr_{(x,s)\sim\mathcal{S}|B_t}[s \leq f_t(x)] \right)^2 \geq \frac{\alpha}{2m} - 22 \sqrt{\frac{12 \left( \ln(\frac{8}{\delta}) + t \ln(4m^2 |\mathcal{G}|) \right)}{n}}.$$

If $n \geq \frac{92928 m^2 \left( \ln(\frac{8}{\delta}) + t \ln(4m^2 |\mathcal{G}|) \right)}{\alpha^2}$, then we have

$$\Pr_{(x,s)\sim\mathcal{S}}[B_t] \left( q - \Pr_{(x,s)\sim\mathcal{S}|B_t}[s \leq f_t(x)] \right)^2 \geq \frac{\alpha}{4m}.$$

As $f_t^*$ achieves the target quantile $q$ against $\mathcal{S}|B_t$ and the quantile error was at least $\frac{\alpha}{4m}$, applying Lemma 3.2 yields

$$PB^{\mathcal{S}}(f_t^*) - PB^{\mathcal{S}}(f_t) \leq -\frac{\alpha}{8m\rho}.$$

**(2) $f_t^* \to \tilde{f}_t^*$:** Recall that $\tilde{\Delta}^*$ results from rounding $\Delta^*$ to the nearest grid point in $[\frac{1}{m}]$. Because $\mathcal{S}|B_t$ is $\rho$-Lipschitz and $f_t(\cdot) + \Delta^*$ satisfies the target quantile $q$ for $\mathcal{S}|B_t$, we can bound the marginal quantile consistency error of $f_t(\cdot) + \tilde{\Delta}^*$ against $\mathcal{S}|B_t$ as

$$\left| \Pr_{(x,s)\sim\mathcal{S}|B_t}[s \leq f_t(x) + \tilde{\Delta}^*] - q \right| = \left| \Pr_{(x,s)\sim\mathcal{S}|B_t}[s \leq f_t(x) + \tilde{\Delta}^*] - \Pr_{(x,s)\sim\mathcal{S}|B_t}[s \leq f_t(x) - \Delta^*] \right| \leq \frac{\rho}{2m}$$

as $|\Delta^* - \tilde{\Delta}^*| \leq \frac{1}{2m}$.

Note that $f_t^*(x) = \tilde{f}_t^*(x)$ for $x \notin B_t$ and $f_t^*(x) = \tilde{f}_{t+1}(x) + (\tilde{\Delta}^* - \Delta^*)$ for $x \in B_t$ where $|\tilde{\Delta}^* - \Delta^*| \le \frac{1}{2m}$. Applying Lemma 3.1 with $\Delta = \tilde{\Delta}^* - \Delta^*$, $\alpha \le \frac{\rho}{2m}$, and $(f, f') = (\tilde{f}_t^*, f_t^*)$, we have that

$$PB_q^{\mathcal{S}}(\tilde{f}_t^*) - PB_q^{\mathcal{S}}(f_t^*) = \Pr_{(x,s)\sim\mathcal{S}}[x \in B_t] \cdot \left( PB^{\mathcal{S}|B_t}(\tilde{f}_t^*) - PB^{\mathcal{S}|B_t}(f_t^*) \right)$$

$$\le \Pr_{(x,s)\sim\mathcal{S}}[x \in B_t] \cdot \left( \frac{\rho}{2m}\frac{1}{2m} - \left(\frac{\rho}{2m}\right)^2 \frac{1}{2\rho} \right)$$

$$\le \frac{\rho}{8m^2}.$$

We have so far shown that with probability $1 - \delta$,

$$PB^{\mathcal{S}}(f_{t+1}) - PB^{\mathcal{S}}(f_t)$$
$$= \left( PB^{\mathcal{S}}(f_{t+1}) - PB^{\mathcal{S}}(\tilde{f}_t^*) \right) + \left( PB^{\mathcal{S}}(\tilde{f}_t^*) - PB^{\mathcal{S}}(f_t^*) \right) + \left( PB^{\mathcal{S}}(f_t^*) - PB^{\mathcal{S}}(f_t) \right)$$
$$\le \left( PB^{\mathcal{S}}(f_{t+1}) - PB^{\mathcal{S}}(\tilde{f}_t^*) \right) + \frac{\rho}{8m^2} - \frac{\alpha}{8m\rho}$$

By union boudning the Chernoff bound (Theorem C.1) over $\mathcal{C}_{t+1}$, we can show that $PB^{\tilde{\mathcal{S}}}(f)$ concentrates around $PB^{\mathcal{S}}(f)$ for every $f \in \mathcal{C}_t$: with probabiliy $1 - \delta$, we simultaneously have

$$\left| PB^{\tilde{\mathcal{S}}}(f_t) - PB^{\mathcal{S}}(f_t) \right| \le \sqrt{\frac{\ln(\frac{2}{\delta}) + t\ln(4m^2|\mathcal{G}|)}{2n}}$$

$$\left| PB^{\tilde{\mathcal{S}}}(\tilde{f}_t^*) - PB^{\mathcal{S}}(\tilde{f}_t^*) \right| \le \sqrt{\frac{\ln(\frac{2}{\delta}) + t\ln(4m^2|\mathcal{G}|)}{2n}}$$

$$\left| PB^{\tilde{\mathcal{S}}}(f_{t+1}) - PB^{\mathcal{S}}(f_{t+1}) \right| \le \sqrt{\frac{\ln(\frac{2}{\delta}) + t\ln(4m^2|\mathcal{G}|)}{2n}}$$

as $f_t, \tilde{f}_t^*, f_{t+1} \in \mathcal{C}_{t+1}$.

In other words, we have with probability $1 - 2\delta$

$$PB^{\tilde{\mathcal{S}}}(f_{t+1}) - PB^{\tilde{\mathcal{S}}}(f_t)$$

$$\le \left( PB^{\tilde{\mathcal{S}}}(f_{t+1}) - PB^{\tilde{\mathcal{S}}}(\tilde{f}_t^*) \right) + \frac{\rho}{8m^2} - \frac{\alpha}{4m\rho} + 4\sqrt{\frac{\ln(\frac{2}{\delta}) + t\ln(4m^2|\mathcal{G}|)}{2n}}.$$

**(3)** $\tilde{f}_t^* \to f_{t+1}$**:** Because $\Delta_t$ is chosen to minimize with respect to the empirical distribution out of grid points $[1/m]$, we can show that the pinball loss against the empirical distribution $\tilde{\mathcal{S}}$ must be lower for $f_{t+1}$ than $\tilde{f}_t^*$.

We can calculate the derivative of the pinball loss with respect to the patch $\Delta$ as

$$\frac{d}{d\Delta} PB^{\tilde{\mathcal{S}}|B_t}(f_t(\cdot) + \Delta)$$
$$= \frac{d\,\mathbb{E}_{(x,s)\sim\tilde{\mathcal{S}}|B_t}[L_q(f_t(x) + \Delta, s)]}{d\Delta}$$
$$= \frac{1}{|B_t|}\frac{d}{d\Delta} \sum_{i:x_i \in B_t} L_q(f_t(x_i) + \Delta, s_i)$$
$$= \frac{1}{|B_t|}\left( \sum_{i:x_i \in B_t, s_i \le f_t(x_i)+\Delta} \frac{d}{d\Delta}L_q(f_t(x_i) + \Delta, s_i) + \sum_{i:x_i \in B_t, s_i > f_t(x_i)+\Delta} \frac{d}{d\Delta}L_q(f_t(x_i) + \Delta, s_i) \right)$$

$$
= \frac{1}{|B_t|} \left( \sum_{i: x_i \in B_t, s_i \leq f_t(x_i) + \Delta} (1 - q) - \sum_{i: x_i \in B_t, s_i > f_t(x_i) + \Delta} q \right)
$$

$$
= \frac{1}{|B_t|} |\{i : x_i \in B_t, s_i \leq f_t(x_i) + \Delta\}| - q
$$

$$
= \Pr_{(x,s) \sim \tilde{\mathcal{S}}|B_t} [s \leq f_t(x) + \Delta] - q.
$$

Because $PB^{\tilde{\mathcal{S}}|B_t}(f_t(\cdot) + \Delta)$ is convex in $\Delta$, minimizing this function is equivalent to minimizing the absolute value of its derivative, which is how $\Delta_t$ is set. Therefore, $PB^{\tilde{\mathcal{S}}|B_t}(f_t(\cdot) + \Delta)$ is minimized at $\Delta_t$. Hence, we have with probability $1 - 2\delta$ ($\delta$ to argue that there is significant quantile consistency error on $B_t$ with respect to $\mathcal{S}$ and $\delta$ to argue that the empirical pinall loss concentrates around its expectation),

$$
PB^{\tilde{\mathcal{S}}}(f_{t+1}) - PB^{\tilde{\mathcal{S}}}(f_t)
$$

$$
\leq \left( PB^{\tilde{\mathcal{S}}}(f_{t+1}) - PB^{\tilde{\mathcal{S}}}(\tilde{f}_t^*) \right) + \frac{\rho}{8m^2} - \frac{\alpha}{8m\rho} + 4\sqrt{\frac{\ln(\frac{2}{\delta}) + t \ln(4m^2|\mathcal{G}|)}{2n}}
$$

$$
\leq \frac{\rho}{8m^2} - \frac{\alpha}{8m\rho} + 4\sqrt{\frac{\ln(\frac{2}{\delta}) + t \ln(4m^2|\mathcal{G}|)}{2n}}
$$

$$
\leq \frac{-\alpha^2}{4\rho^3} + 4\sqrt{\frac{\ln(\frac{2}{\delta}) + t \ln(4m^2|\mathcal{G}|)}{2n}}
$$

as we have chosen $m = \frac{\rho^2}{2\alpha}$.

If $n \geq \frac{512\rho^6 \left( \ln(\frac{2}{\delta}) + t \ln(4m^2|\mathcal{G}|) \right)}{\alpha^4}$, we have

$$
PB^{\tilde{\mathcal{S}}}(f_{t+1}) - PB^{\tilde{\mathcal{S}}}(f_t) \leq -\frac{\alpha^2}{8\rho^3}.
$$

Because we decrease the empirical pinball loss by $\frac{\alpha^2}{8\rho^3}$ in each round with probability $1 - 2\delta$, we have that with probability $1 - 2\delta \cdot \frac{8\rho^3}{\alpha^2}$, the algorithm halts at round $T = \frac{8\rho^3}{\alpha^2}$. If

$$
n \geq 92928 \left( \ln \left( \frac{8}{\delta} \right) + \frac{8\rho^3}{\alpha^2} \ln(4m^2|\mathcal{G}|) \right) \max \left( \frac{m^2}{\alpha^2}, \frac{\rho^6}{\alpha^4} \right),
$$

we satisfy all the requirements that we stated previous for $n$ in each round $t \in [T]$.

If we set $\delta' = \frac{16\rho^3}{\alpha^2}\delta$, we have with probability $1 - \delta'$, the algorithm halts in $T = \frac{8\rho^3}{\alpha^2}$ where we require

$$
n \geq 92928 \left( \ln \left( \frac{128\rho^3}{\alpha^2\delta'} \right) + \frac{8\rho^3}{\alpha^2} \ln(4m^2|\mathcal{G}|) \right) \max \left( \frac{m^2}{\alpha^2}, \frac{\rho^6}{\alpha^4} \right)
$$

$$
= 92928 \left( \ln \left( \frac{128\rho^3}{\alpha^2\delta'} \right) + \frac{8\rho^3}{\alpha^2} \ln \left( \frac{\rho^4|\mathcal{G}|}{\alpha^2} \right) \right) \max \left( \frac{\rho^4}{4\alpha^4}, \frac{\rho^6}{\alpha^4} \right)
$$

$\square$

## D  ADDITIONAL EXPERIMENTS AND DISCUSSION

### D.1  A DIRECT TEST OF GROUP CONDITIONAL QUANTILE CONSISTENCY AND QUANTILE MULTICALIBRATION

In this section we abstract away the non-conformity score, and directly perform a direct evaluation of the ability of `BatchGCP` and `BatchMVP` to offer group conditional and multivalid quantile

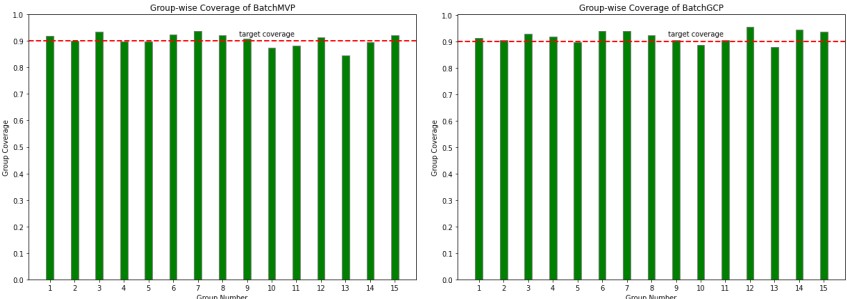

Figure 6: Per-group coverage of `BatchMVP` (left) and `BatchGCP` (right) on a representative run.

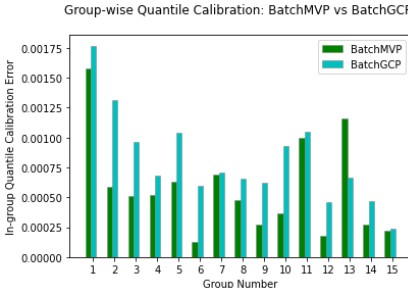

Figure 7: Per-group calibration error of `BatchMVP` and `BatchGCP` on a representative run.

consistency guarantees. We produce a synthetic regression dataset defined to have a set of intersecting groups that are all relevant to label uncertainty. Specifically, the data $\{(x_i, y_i)\}_{i=1}^{10000} \in (\mathbb{Z}_+ \times \mathbb{R})^{10000}$ is generated as follows. First, we define our group collection as $\mathcal{G} = \{g_1, \ldots, g_{15}\}$, where for each $j = 1, \ldots, 15$, the group $g_j = \{j, 2j, 3j, \ldots\} \subset \mathbb{Z}_+$ contains all multiples of $j$. Note that group $g_1$ encompasses the entire covariate space, ensuring that our methods will explicitly enforce *marginal* coverage in addition to group-wise coverage. Each $x_i$ is a random integer sampled uniformly from the range $[1, 5000]$. We let the corresponding label $y_i = \frac{|y_i'|}{|y_i'|+1} \in [0, 1]$, where $y_i'$ is distributed as the sum of $n(x_i)$ i.i.d. $\mathcal{N}(0, 1)$ random variables, where $n(x_i)$ is the number of groups that $x_i$ belongs to. After generating our data, we split it into 80% training data $\mathcal{D}_{train}$ and 20% test data $\mathcal{D}_{test}$. We then run both methods on the group collection $\mathcal{G}$, for target coverage $q = 0.9$, with $m = 100$ buckets.

The group coverage obtained by both methods on a sample run can be seen in Figure 6. Generally, both methods achieve target group-wise coverage level on all groups, with `BatchMVP` exhibiting less variance in the attained coverage levels across different runs. Meanwhile, the group-wise quantile calibration errors of both models are presented in Figure 7. Both `BatchMVP` and `BatchGCP` are very well-calibrated on all groups, with calibration errors on the order of $10^{-3}$ — even though unlike `BatchMVP`, the definition of `BatchGCP` does not enforce this constraint explicitly.

### D.2    FURTHER COMPARISONS ON STATE CENSUS DATA

In Section 5.2 we performed an income-prediction task using census data (Ding et al. (2021)) from the state of California to compare the performance of `BatchGCP` and `BatchMVP` against each other, as well as against other regularly used conformal prediction methods. Here, we present results of the same experiment using data from other US states. We selected the ten largest states (by population data) to work with, of which California is one. Results for every state are averaged over 50 runs of the experiment, taking different random splits over the data (to form training, calibration, and test sets) each time. The plots in Figure 8, Figure 9, Figure 10 and Figure 11 compare the performance of all four methods with metrics such as group-wise coverage, prediction-set size, and group-wise quantile calibration error.

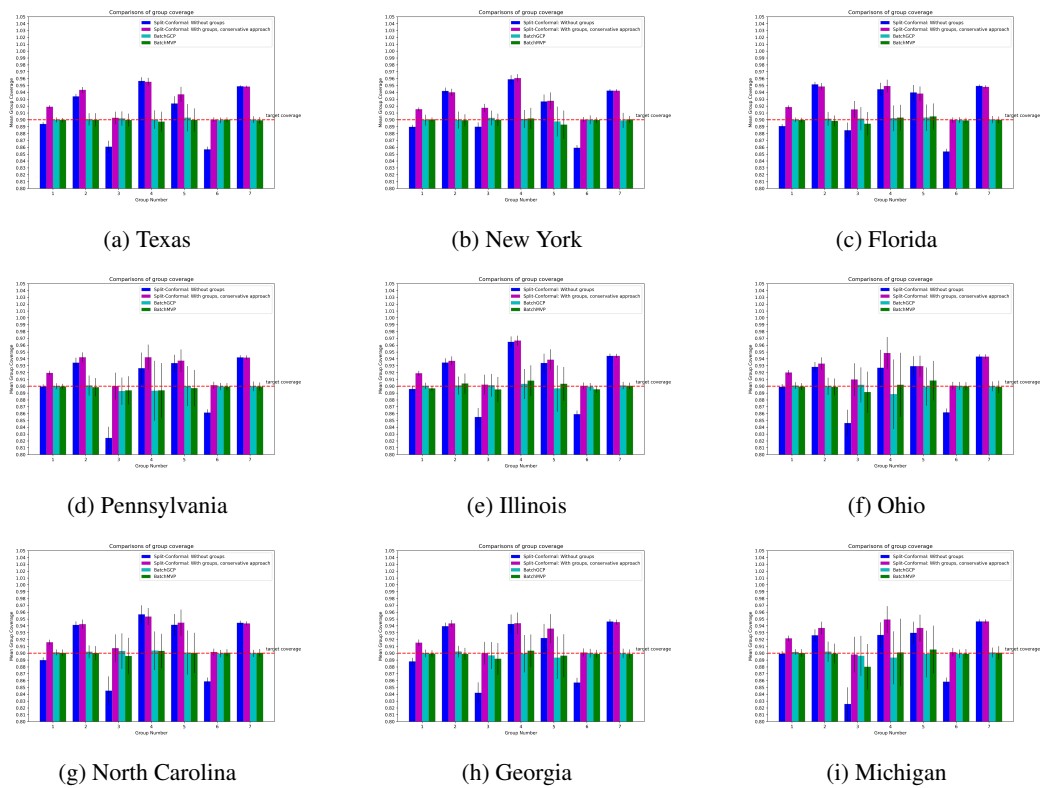

Figure 8: Results for group-wise coverage for nine different states (averaged over 50 runs) using different methods of conformal prediction. Error bars show standard deviation.

### D.2.1 HALTING TIME

Recall that our generalization theorem for `BatchMVP` is in terms of the number of iterations $T$ it runs for before halting. We prove a worst-case upper bound on $T$, but note that empirically `BatchMVP` halts much sooner (and so enjoys improved theoretical generalization properties). Here we report the average number of iterations $T$ that `BatchMVP` runs for before halting on each state, averaged over the 50 runs, together with the empirical standard deviation.

Texas: $10.84 \pm 1.2059$.
New York: $11.26 \pm 1.2134$
Florida: $12.06 \pm 1.4752$
Pennsylvania: $11.68 \pm 2.3189$
Illinois: $9.74 \pm 1.4941$
Ohio: $10.94 \pm 1.7482$
North Carolina: $13.22 \pm 1.5138$
Georgia: $12.18 \pm 1.6454$
Michigan: $11.24 \pm 2.0353$

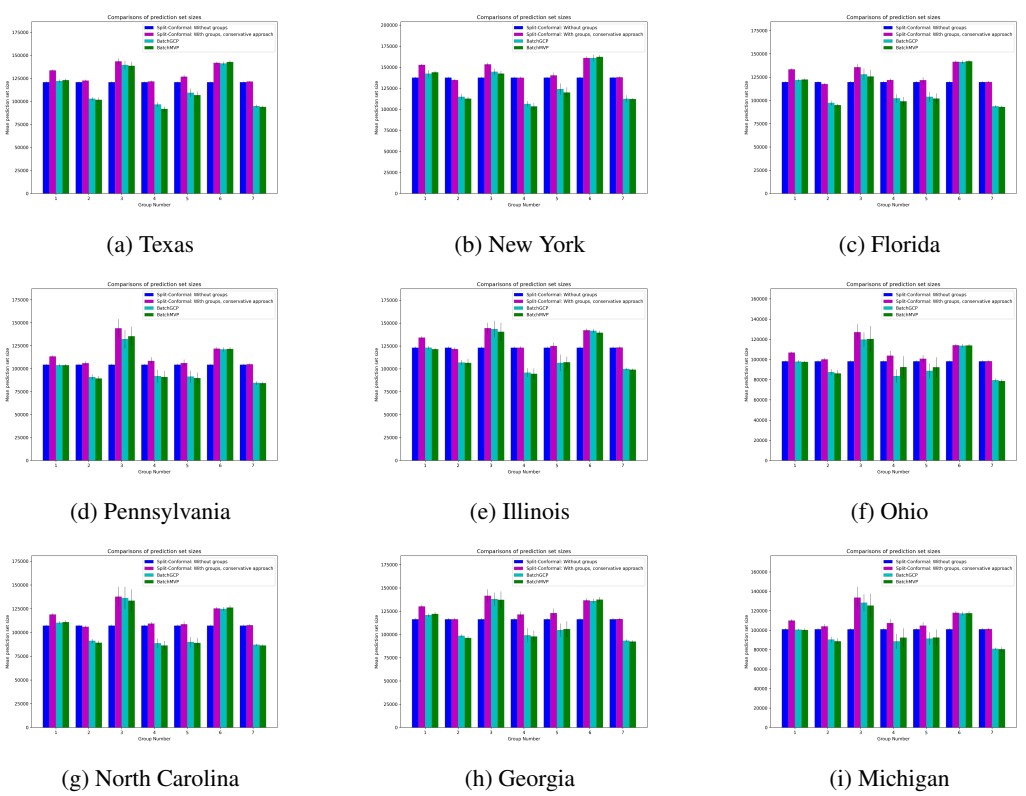

Figure 9: Results for group-wise average prediction set size for nine different states (averaged over 50 runs) using different methods of conformal prediction. Error bars show standard deviation.

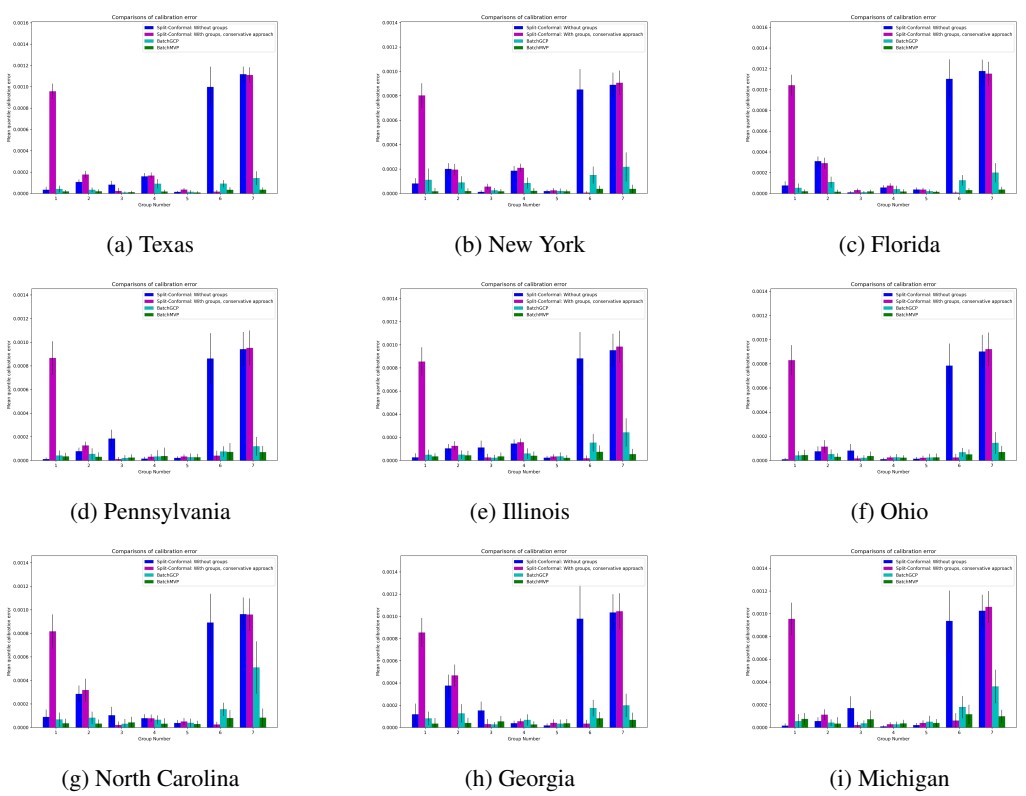

Figure 10: Results for group-wise calibration error (averaged over 50 runs). Error bars show standard deviation.

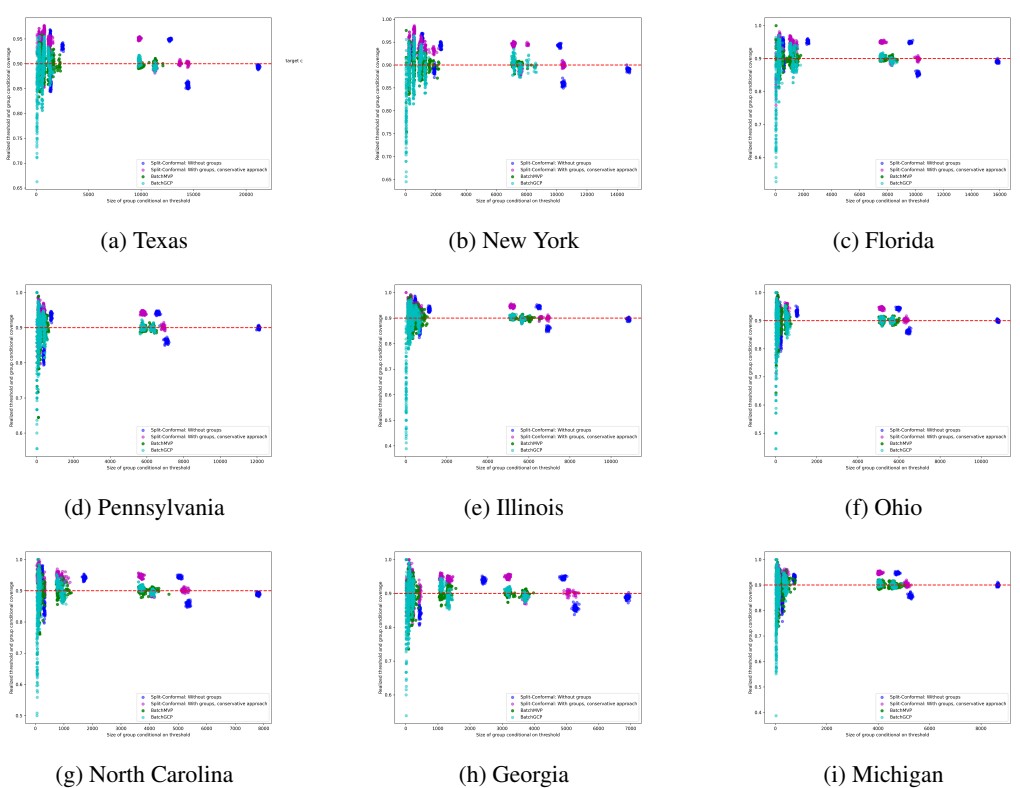

Figure 11: Scatterplots of the number of points associated with each threshold-group pair $(g, \tau_i)$ against the average coverage conditional on that pair for all $g \in \mathcal{G}$ and all $\tau_i$ in a grid, over all tested conformal prediction methods (consolidating results over all 50 runs), for all nine states. Target coverage is $q = 0.9$.

