# OpenReview forum: "Batch Multivalid Conformal Prediction"
_ICLR.cc/2023/Conference — ICLR 2023 poster_

### Official Review · Reviewer_auqC · 2022-10-25

**Confidence:** 3
**Clarity, Quality, Novelty And Reproducibility:** See above.
**Correctness:** 3
**Technical Novelty And Significance:** 3
**Empirical Novelty And Significance:** 3
**Recommendation:** 8

**Strength And Weaknesses:**

Strength:

Theoretical analysis of the proposed method explains why it works.

Adequate experiments are conducted.

Weaknesses:

-- The first sentence in the last paragraph on page 3, "that on each input x output a value f(x) that is", output should be outputs.

--Please provide some examples that satisfy the Lipschitz condition in Definition 2.2.

-- what is the meaning of this sentence "To facilitate learning models f with guarantees conditional on their output values, ..."?  Please give some examples of such f. Is m here the same as m in Algorithm 2? In general regression settings, finite cardinality seems unrealistic. In Section 5, please state clearly what f you have used, does it satisfy the finite cardinality condition? How do you choose m in practice (in Section 5, you use different m)?

--From theoretical results, the sample size should be large enough for the results to hold (with high probability). For a comprehensive understanding of the sample size requirement, It will be helpful to provide more experimental results for different sample sizes (especially for small sample sizes).

-- Is it possible to theoretically analyse the (average) size of predictive sets of the proposed algorithms (smaller compared to previous conformal methods)? Please provide some discussions.

-- Please explain the parameters alpha, rho, m in Algorithm 2 and their connection. Are these parameters pre-specified by users or tuned in some way?

-- The meaning of Figures 2 and 4 is not very clear. Please clarify this.

-- From experimental results (e.g., Figures 1&3), the method, BatchMVP, might obtain predictive sets that undercover in some groups. Can you explain this?

**Summary Of The Paper:**

This paper proposes a conformal prediction algorithm that obtains multivalid converge on exchangeable data in the batch setting. Multivalid coverge guarantees mean that the target coverage level holds conditionally on membership in each group or/and the threshold value. They provide theoretical analysis to clarify the reliability of the proposed algorithms under some regularity conditions. Extensive experiments have been conducted to compare the performance of the proposed method and other comparative methods.

**Summary Of The Review:**

This paper proposes a novel method that has multivalid coverage guarantees. They theoretically analyse the properties of the proposed method.

---

> ### Author Response · Authors · 2022-11-09
> **Response to Reviewer auqC. Part 1**
>
> Thank you very much for your thoughtful feedback. We have fixed the typos you identified; Below we address your substantive questions.
>
> (1) Examples of lipschitz distributions\
> Any continuous distribution with bounded densities satisfies our Lipschitz assumption --- so most standard distributions (e.g. the Gaussian distribution) satisfy it. But more to the point, -any- distribution that is perturbed by continuous bounded noise (e.g. the uniform distribution in $[-\epsilon,\epsilon]$) will satisfy it --- so if we want, we don’t have to assume that the non-conformity score distributions satisfies this condition, we can explicitly enforce it by perturbing our non-conformity scores with small amounts of noise.
>
> (2) Learning $f$ with finite values \
> Our algorithm BatchMVP explicitly learns a model f whose output range takes on one of at most $m+1$ discrete values. Note that we are not assuming anything about the true quantiles of the distribution (which need not lie on this discrete grid) --- rather this is a design choice that we make so that we can bound the complexity of the models that we learn which is important for proving out-of-sample generalization bounds.
>
> In Theorem 3.2, m is set to be $8\rho^2/\alpha$. As we state in our experimental section, in all of our experiments we use m = 100, and we do not vary this across experiments.
>
> It might be helpful to note that previous conformal prediction methods learn a single threshold $\tau$ (or equivalently, a constant function $f(x) = \tau$) --- and so prior methods have “$m$” set to 1. We increase “$m$” in our work to be able to learn a more expressive model that gives correspondingly strong guarantees.
>
>
>
> (3) Sample size
> We agree that it is interesting to understand how the generalization error varies with sample size. We ran an extensive set of experiments and observed strong generalization performance across all of them, but we did not explicitly investigate generalization error as a function of sample size; we will consider adding such an experiment to the paper.
>
>
> (4) Size of the prediction sets
> Following the literature on conformal prediction, we give a wrapper algorithm that can be used together with any non-conformity score s. In general it is difficult to say anything about the size of the prediction sets in such a general setting, because they depend heavily on the choice of non-conformity score. When compared to methods that offer conservative groupwise coverage guarantees using the same non-conformity score (one of our benchmarks), we are guaranteed to produce smaller average prediction set sizes, because coverage rates vary monotonically with the threshold chosen (which directly determines the prediction set size). When we compare to naive methods that ignore groups and offer only marginal coverage (another of our benchmarks) we will tend to offer smaller prediction set sizes on those groups for which the naive method over-covers, and (correctly) offer larger prediction set sizes on those groups for which the naive method under covers. Our experiments bear this out.
>
> (5) $(\alpha, \rho, m)$ \
> $\alpha$ specifies the approximation guarantee the algorithm is aiming for, and determines the halting condition of the algorithm. $m$ is the discretization parameter that determines the range of the model that we learn. $\rho$ is the Lipschitz parameter on the non-conformity score distribution and is not directly used by the algorithm. To prove our bounds, we set $m = 8\rho^2/\alpha$. So in theory, there is only one parameter to set --- the target accuracy $\alpha$. $\rho$ is a problem dependent constant, and $m$ is set as a function of the other parameters. In practice, in all of our experiments, we fix $m = 100$.

---

> > ### Author Response · Authors · 2022-11-09
> > **Response to Reviewer auqC. Part 2**
> >
> >
> > (6) Meaning of figure 2 and 4 \
> > Figure 2 shows that our prediction sets are quantile multicalibrated. The first histogram plot shows what the weighted quantile calibration error is for each group where quantile calibration error is defined in Definition 3.4, and the error is weighted by the actual group size meaning we’ve multiplied the quantile calibration error by $\Pr[g(x) = 1]$ for each group $g$. The second plots are scatter plots where each point corresponds to a (group, threshold) pair. The x-axis records the number of data points corresponding to this (group, threshold) pair, and the y-axis records the coverage rate on points corresponding to this (group, value) pair. These plots are consistent with our theory: the larger the group sizes, the closer the average coverage is to the target coverage. We will clarify these plots.
> >
> > (7) Undercovering by BatchMvp \
> > The traditional method for split conformal prediction offers one-sided coverage guarantees by explicitly over-covering on the calibration set by an amount equal to the expected coverage error, which results in expected out of sample coverage error that is always at least the target $q$ and not “too much” larger than $q$, where “too much” is a function of the number of samples $n$. We offer group-wise guarantees, where (necessarily) the coverage error on a group $g$ depends not on $n$ (the total number of samples), but on the number of samples we have observed from group $g$. Since individual data points can be members of multiple groups (both large and small), it is not possible to promise one-sided coverage guarantees by uniformly over-covering on the calibration set, because the coverage error is not uniform across different groups. So, we (necessarily) offer two sided coverage guarantees: the coverage on any particular group is guaranteed to be close to the target $q$ (where close is a function of the size of the group), but might either over or under cover. However, as our experiments demonstrate, our coverage is very good empirically, and has group-wise under coverage that is substantially less than the standard method for split conformal prediction which ignores group structure.
> >
> > Thanks once again for your detailed set of questions. If you have any other questions, we’ll be happy to answer them!

---

### Official Review · Reviewer_3CEp · 2022-10-27

**Confidence:** 2
**Correctness:** 4
**Technical Novelty And Significance:** 3
**Empirical Novelty And Significance:** 3
**Recommendation:** 6

**Clarity, Quality, Novelty And Reproducibility:**

Can do better in terms of clarity, see my remarks above and below.

I could not check the proofs given the time delay, and length of the paper.


**Strength And Weaknesses:**

The idea of learning a threshold function is very innovative and can certainly be applied in other contexts. The numerical demonstration is very convincing and the subject in general is relevant for the AI/ML community.

The major weakness of the paper is that it is very hard to read. The proof does not seem to be written to be read at all. Same expressions are repeated several times, making it very messy, and very difficult to understand. The authors would benefit from reducing this clutter in the proofs. An example is the proof of Lemma C4, where the core tool is just the application of classical remarkable identities to bound $|(q - a)^2 - (q - b)^2 |$.

Also, it is very difficult to understand under which hypotheses the proposed algorithms work, so much the article is convoluted with additional (sub)-constraints on the distributions. And this, from lemma to lemma, one gets lost and even after reading it multiple times, I have trouble extracting a clear setting where the algorithms are guaranteed to work.

**Summary Of The Paper:**

This paper extends the scope of conformal prediction techniques from marginal validity to conditional validity with respect to a subset of the feature space. This allows for example to quantify the uncertainty of predictions conditional on the race or gender of the subpopulations. The main novelty lies in the fact that instead of thresholding the score functions by a constant to satisfy the prescribed coverage, the authors propose to learn a threshold function under coverage validity constraints. This is very similar to what is done in learning under fairness constraints. The authors present a series of numerical experiments illustrating the performance of their method, a clear improvement over the alternative method.

**Summary Of The Review:**

Some comments/questions:

1) By construction, Conformal Prediction is distribution free. The propositions of this paper are not, since they significantly restrict the CDF with some regularity assumptions, that in-fine, cannot be verified because the distribution of the data is unknown. Can the authors comments on that?

2) A core assumption, all around the paper, is that the scores functions are $\rho$-Lipschitz. However, the latter depends on the distribution, which is also unknown, Even more, the scores are constructed from the data eg as prediction errors of a neural net. How do one verify this assumption? More annoying, the proposed algorithms do not even use the regularity constants in play, which suggests, with a bitter taste, that the theoretical analysis is not connected to the proposed algorithm. The authors might want to discuss this a bit more.

3) Considering the context of the article, the exchangeability assumption is perhaps a bit too strong. It would be surprising if the population of the considered subgroups follow the same distribution nonetheless being independent. When considering age or gender, can the authors present any example where such assumption is realistic?

4) When conditioning on an under-represented group (ie where very low number or none of examples is observed), the conditional event becomes a rare event. How does it affect the length of the conformal set proposed?

---

> ### Author Response · Authors · 2022-11-09
> **Response to Reviewer 3CEp**
>
> Thank you for your careful reading! We’ll try to answer your questions and clear up confusions.
>
>
> (1) The Lipschitzness Assumption on the Distribution\
> Our convergence analysis for BatchMVP (Theorem 3.2) requires the assumption that the non-conformity score distribution is Lipchitz. This is a mild assumption that can be explicitly enforced if desired --- for example by perturbing the non-conformity scores with noise from a continuous distribution (for example uniformly random noise in $[-\epsilon,\epsilon]$. This is already standard practice in conformal prediction --- for example, it is an option in the standard conformalized quantile regression (CQR) package --- but in fact in our experiments we find that this is not needed. We will add more explicit discussion of this.
>
> The lipschitzness condition -does- come into the design of our algorithm --- in particular, we set our discretization parameter $m$ in the algorithm in terms of $\rho$ in Theorem 3.2.  Note that it is not possible to prove the kinds of bounds that we prove without any assumption on the distribution (since we prove bounds that guarantee both that we do not substantially under-cover -and- that we do not substantially over-cover). To see this, consider a distribution on non-conformity scores that is just a point mass --- i.e. there is only a single non-conformity score. The only possible threshold calibrated coverage rates are 0% and 100%. To hit target coverage rates in between, it is necessary to assume that the non-conformity score distribution is continuous, and the Lipschitzness parameter controls our rate of convergence.
>
>
> (2) Exchangeability  \
> We want to clarify a misconception that we think the reviewer has about the exchangability assumption. This just means that the overall data distribution is permutation invariant --- it *does not* imply that the distribution of features conditional on group membership are the same. So for example, in our Census experiments, we get exchangability simply because we have partitioned the training and test data at random --- which does not mean that the data is distributed identically as a function of gender, race, etc. In fact we see in our experiments that it is not, which is why standard split conformal prediction methods have coverage rates that differ by group. It is exactly because the group-wise distributions differ that methods like ours are necessary.
>
>
> (3) Conditioning on a small groups \
> As you correctly observe, it is impossible to provide strong bounds for groups that are too small, since we will not have observed sufficiently many samples from those groups (for groups that have probability < $1/n$, where n is the number of datapoints we have, potentially no samples at all!). This is reflected in our theorems: our alpha-approximate coverage guarantees when applied to a group g are normalized by sqrt{probability mass of the group} --- see, for example, Theorem C.3 for BatchGCP which bounds the coverage error on group $g$ by $\sqrt{\alpha}/\sqrt{Pr[g(x) = 1]}$. Thus the bounds that follow from our theory smoothly degrade for smaller group sizes --- as they must.
>
> Thanks once again for your detailed set of questions --- if you have others, we’ll be happy to answer them!

---

> > ### Comment · Reviewer_3CEp · 2022-12-02
> > **Clarity and Lipschitz**
> >
> > Thank you for the clarifications and precisions provided.
> > As a last comment, it would be nice to add the explanations about the lipschitz hypothesis on the paper. Furthermore, it would be nice to explicitly provide the appropriate constants on some examples, and discuss a potential trade-off that would result from the addition of random noise. Intuitively, this would be necessary as it is in differential privacy.
> >
> > Finally, any improvement of the writing of the proofs is totally welcome. The article contains technical contributions on a challenging problem and so it will be interesting for some people to understand the details of the proof.

---

> > > ### Author Response · Authors · 2022-12-02
> > > **Thanks for the comment; we'll implement it!**
> > >
> > > Thanks --- your feedback on how to improve the clarity of the paper is welcome, and we will incorporate it into our paper. We can no longer upload revisions to the paper since the official discussion period has ended, but if the paper is accepted we will incorporate a fuller discussion of the Lipschitz assumption and how it can be enforced with noise into the camera ready. We will also work to make the proofs as clear as possible without sacrificing rigor.

---

### Official Review · Reviewer_2Lnk · 2022-10-27

**Confidence:** 3
**Correctness:** 3
**Technical Novelty And Significance:** 3
**Empirical Novelty And Significance:** 3
**Recommendation:** 8

**Clarity, Quality, Novelty And Reproducibility:**

The paper is nicely written however there are certain parts when it comes to intuition on what the theorems mean that are still none intuitive to me. The experiments are quite well laid out with the exception of the questions raised above.
Overall the novelty is there and I quite like the angle they tackle the problem. The experimental results also back up their claims however I have not run the code myself.

**Strength And Weaknesses:**

I will start with the strengths of this paper:
- The authors propose a novel way to understand batch conditional coverage by utilizing the pinball loss in a specific way.
- They also analyse the fact that the pinball loss might not always be well optimized and hence provide an iterative algorithm that sequentially improves the quantile computation.
- They theoretically analyze the proposed algorithms and provide further theoretical analysis of how their method can achieve the guarantees (I am not familiar with the theory and hence I will be leave this part to be scrutinized by the other reviewers)
- The experiments are mostly convincing and show that their proposed method is superior to current baselines

In terms of weaknesses and clarifications:
- First of all I would like the authors to clarify what $f(x) = \tau$ actually signifies. I am a little confused because wouldn't that mean f is a constant function? I just don't get the intuition on what the point of $\tau$ is. Could the authors please give a concrete example of what tau signifies and why f(x) is constant?
- In algorithm 1, I am confused to why you are computing the pinball loss wrt to y shouldn't it be the scores? maybe I am completely misunderstanding this part here but isn't f(x) supposed to be the quantiles for the scores? in that case, why are we minimizing the loss wrt to y the output? Sorry if I misunderstood sth.
- I would suggest the authors to create a figure on how the patching works on a simple 1-2D example. I think that would illustrate the algorithm much easier. I get the general gist. My confusion also comes into algorithm 2. Isn't g(x) supposed to be fixed? i.e. these are the subgroups that we predefined. Now that we are optimizing over them how can we ensure that the groups that come out at the end reassemble the groups that we are interested in dealing with when all training is done? Also, what is the form of g?
-  I am not familiar with PAC Bayes but a constant that big in 90k+ in theorem 4.2 seems not to be very useful. Even if it works in practice the theorem seems kind of useless. Please correct me if I am wrong here.
- Could you please point me to the experiments where you increased the group size? I am confused to what breaks in your theory if each group is just an individual datapoint i.e. each group is just a 1-2 datapoint. Is there something that breaks in your theory or does that mean the bounds will become useless? I am just trying to get a more intuitive understanding on how the group size affects the overall bounds as in standard CP.
- This is more like a clarification question: In the Barber et al 2020 paper do they do the following: split the calibration data for each subgroup and then compute a different threshold for each group. Then for testing time, they first check which group it is in and then sue the corresponding threshold for that new datapoint. If the answer is yes then please just reply with a yes, if no, please tell me why this is a  valid baseline if the calibration data is reasonably big for each group.


**Summary Of The Paper:**

The paper looks at a novel way to investigate batch conditional coverage in the setting of conformal prediction. CP is known to be unable to provide conditional coverage guarantees and hence this paper proposed two methods to relax these assumptions but taking a closer look at batch conditional coverage and hence propose batchGCP and batchMVP.

The key insight is that their threshold contrary to CP is now a function of X which in turn is now to be learnt.

By adding this extra degree of freedom they are able to obtain interesting results regarding batch conditional coverage and show that their proposed method is superior compared to standard CP and other baselines.

**Summary Of The Review:**

I have detailed all my concerns above and I am happy to increase my score if the above misunderstandings have been clarified.

---

> ### Author Response · Authors · 2022-11-09
> **Response to Reviewer 2Lnk. Part 1**
>
> Thanks for your thoughtful feedback. We’ll try to answer your questions and clarify confusions/concerns.\
>
> (1) $f(x) = \tau$\
> Traditional methods of split conformal prediction choose a single threshold $\tau$, which defines prediction sets $P(x) = \\{y : s(x,y) \le \tau\\}$. We choose a functional representation instead, using prediction sets $P(x) = \\{y : s(x,y) \le f(x)\\}$. So traditional methods would use a constant function in this notation, but we do not.  For instance, BatchGCP outputs f(x) whose value is determined by the groups that point x belongs to.  When we include the conditioning event $f(x) = \tau$ in our coverage guarantees for particular values of $\tau$, this means that our coverage guarantees have to hold not just overall, but also on the subset of the data on which the function $f(x)$ happens to take a particular value $\tau$. This is an important guarantee, because e.g. it rules out a solution that sets $f(x)$ to its maximal value with probability $q$ and $f(x)$ to its minimal with probability $(1-q)$. This obtains the target marginal and group-wise coverage rate, but does not obtain it conditional on the value of $f(x)$.
>
>
> (2) Calculating the pinball loss in algorithm 1 \
> You are right --- this is a typo --- thanks for catching it! We should be computing the pinball loss with respect to the scores $s(x,y)$, not with respect to y (which might not even be numeric!) We have corrected the typo.
>
> (3) Confusion about algorithm 2 \
> We make no assumption about the collection of groups $G=\\{g_1, … , g_k\\}$: they are just arbitrary subsets of X and hence can intersect, and they do not change over the course of the algorithm. We represent groups with their indicator functions: so $g(x) = 1$ means that $x$ is in group $g$, and $g(x) = 0$ means that x is not in group $g$.
>
> Algorithm 2 is given some collection of groups as an input. In each round $t$, whenever $f_t$ is not $\alpha$-approximately quantile multicalibrated, we find group $g \in G$ and $v \in [\frac{1}{m}]$ for which $\Pr[g(x) = 1, f_t(x) = v] (q - Pr[s \leq f_t(x) | f_t(x) = v, g(x) = 1])^2$ is maximized and name this pair $(g_t, v_t)$. Note that $(g,v)$ is not fixed and is the variable over which we take the argmax. We’ll include an example and include a few iterations of the algorithm on the example if confusion remains.
>
> (4) Big constant in theorem 4.2\
> Note that we give -two- generalization theorems. Theorem 4.1 does not have a large constant, and bounds the generalization error of our algorithm as a function of the number of rounds T it takes to converge. Theorem 4.2 (the one with the large constant) proves a worst-case upper bound on the number of rounds the algorithm must take to converge. But Theorem 4.1 can be directly applied using the number of rounds the algorithm takes to converge empirically, which we record for all of our experiments and is generally small --- for example we note that the average number of iterations for convergence is only 10.64 in the income prediction experiments we report in the main body. We further note that all of our experimental results are reported on held-out data, confirming that our method has strong out of sample performance.
>
>
>
> (5) Group size\
> As you correctly observe, it is impossible to provide strong bounds for groups that are too small, since we will not have observed sufficiently many samples from those groups (for groups that have probability < 1/n, where n is the number of datapoints we have, potentially no samples at all!). This is reflected in our theorems: our alpha-approximate coverage guarantees when applied to a group g are normalized by sqrt{probability mass of the group} --- see, for example, Theorem C.3 for BatchGCP which bounds the coverage error on group g by $\sqrt{\alpha}/\sqrt{Pr[g(x) = 1]}$. Thus the bounds that follow from our theory smoothly degrade for smaller group sizes --- as they must.
>
> Experimentally we investigate the degradation of our bounds with group size in the right hand side plots of figures 2 and 4; we have more of these style of plots in the Appendix. In these plots, each point corresponds to a (group, threshold) pair, the x-axis is the number of points in this set (i.e. the number of points in the group such that $f(x)$ takes value equal to a particular threshold $\tau$), and the y-axis is the average coverage over this set of points. The results are consistent with our theory: coverage is closer to the target value for larger groups of points (those farther to the right on the plot)

---

> > ### Author Response · Authors · 2022-11-09
> > **Response to Reviewer 2Lnk. Part 2**
> >
> > (6) Comparison to Barber et al.
> > What you describe is what Barber et al. 2020 (and Romano et al 2020a) do when the groups are disjoint. When the groups are not disjoint (as in our case), it is not well defined to use “the corresponding” threshold, since an example that is a member of multiple groups will correspond to multiple thresholds. What they do in this case is use the largest (most conservative) threshold out of all of the thresholds corresponding to groups the example is a member of. We compare to this benchmark in all of our experiments under the name “Split conformal: With groups, conservative approach”. As expected it generally leads to over-coverage and larger than necessary prediction sets.
> >
> > Thanks for your detailed review --- we’re happy to answer other questions you may have. We hope our answers were clarifying!

---

> ### Author Response · Authors · 2022-11-20
> **A gentle ping**
>
> Hi! Now that the discussion period has ended, we thought we would send a gentle ping (we know this is a busy time of year). If we have addressed your questions to your satisfaction, we would appreciate it if you could raise your score as you graciously have offered to do! If not, we are happy to answer any remaining questions that you have.

---

> > ### Comment · Reviewer_2Lnk · 2022-12-11
> > **Response**
> >
> > First of all thanks for the rebuttal and sorry for the late reply as I had fallen sick in recent weeks and am only back to normal now ...
> >
> > I have read the rebuttal and am happy with the response.
> >
> > I have raised my score accordingly.
> >
> > Best

---

> > > ### Author Response · Authors · 2022-12-11
> > > **Thanks**
> > >
> > > Thanks --- sorry to hear about your illness, and glad that you are feeling better.

---

### Official Review · Reviewer_FUsZ · 2022-11-03

**Confidence:** 3
**Clarity, Quality, Novelty And Reproducibility:** Hard for me to understand parts of it.
**Correctness:** 3
**Technical Novelty And Significance:** 3
**Empirical Novelty And Significance:** 3
**Recommendation:** 6

**Strength And Weaknesses:**

Strength. This paper is technically sound. Conditional on group membership has important application.

Weakness. I am a bit confused about the writing. I am generally familiar with a conformal prediction on the statistics side, but this paper is a bit hard for me to parse. I think it mainly follows a line of literature in CS, which I am not familiar with, and a terminology explanation is missing. For instance, I cannot understand what is the difference between the batch setting in this paper and the setting studied in standard literature, though the paper clearly says ``it is different from the sequential setting because the labels are not applicable when the set is deployed".  The algorithms are based on s, which involves y, does it mean they can only observe s and x? And the sequential setting I think it means online learning, but most of the standard conformal settings are just offline, what is the difference between those literature and batch settings? I also could not understand why a criterion needs to hold for all tau when conditional on f(x)=tau, this is a bit too strong. Also by replacing tau to f(x) is not novel, it is widely considered in the literature such as ``conformal quantile regression" by Romano. Besides, the lipchitiz on distribution basically needs the density to be bounded, which is not applicable on many real examples in practice, such as applications in learn then test paper by Bates etc.

Some minor comments, the set of distribution on \mathcal X should use \Delta_{\mathcal X} instead of \Delta \mathcal X, which looks weird.

I would raise my score if the author can provide more detailed explanation.

**Summary Of The Paper:**

This paper studies distribution-free multi-valid coverage in the batch setting. They provide two algorithms to achieve multi-valid coverage, which is a stronger notion than regular marginal coverage. In a brief summary, they design their algorithm based on a theoretical argument that patch operation such that a postprocessing and make sure coverage guarantee when conditional on x\in B  for a set B can decrease pin-ball loss, which is the a key loss for quantile control.

**Summary Of The Review:**

See commentes above.

---

> ### Author Response · Authors · 2022-11-09
> **Response to Reviewer FUsZ**
>
> Thanks for your thoughtful feedback. We’ll try to answer your questions and clarify confusions.
>
> (1) The batch vs. sequential setting. \
> What we call the “batch” setting is the traditional setting for conformal prediction, in which one typically assumes there is an exchangeable data distribution. A sample of labeled data is assumed to be available to train the model and calibrate a threshold on the non-conformity score (e.g. in split conformal prediction, the labeled data is partitioned into a training and calibration set).  Then at test time, new unlabeled data arrives, and using our model and non-conformity score threshold, we affix the new data with a prediction interval. This does not require labels at test time, and comes with guarantees assuming the data is exchangeable. This standard setting is the one we work in.
>
> What we call the sequential setting is the (less commonly studied) setting in which the learner sequentially encounters feature $x_t$, must produce prediction sets, and then observes the label $y_t$. In this setting it is possible to give coverage guarantees without assuming exchangability (or indeed anything) about the data distribution --- but this setting requires that the labels be observed at test time after making predictions. This is the setting studied e.g. by Gibbs and Candes in “Adaptive Conformal Inference Under Distribution Shift”. The reason we make this distinction is that previous work studying multivalid coverage in conformal prediction (Bastani et al. “Practical Adversarial Multivalid Conformal Prediction” and Gupta et al. “Online Multivalid Learning”) operates in the sequential setting. Our main contribution is giving algorithms with multivalid guarantees in the more standard batch setting, in which we require exchangability, but do not require having access to labels at test time.
>
> (2) Conditioning on $f(x)$ \
> We give two algorithms --- one of which asks for coverage only conditional on group membership (group conditional coverage), and the other of which asks for the stronger guarantee of coverage conditional on group membership and the threshold $f(x)$ (multivalid coverage). In general, the reason to want to condition on more is to get closer to the ideal of full conditional coverage: the prediction set should have guarantees over the randomness only of the label distribution, not over the feature vector. Group conditional coverage gets closer to this compared to the standard guarantee of marginal coverage, but it is still vulnerable to the following kind of behavior: Consider a predictor that on a uniformly random 95% of examples predicts the full (trivial) prediction set (by setting $f(x)$ to take on its maximal value), and on 5% of examples predicts the empty prediction set (by setting $f(x)$ to take on its minimal value). It obtains 95% group-wise coverage, but any particular prediction set has either 100% or 0% coverage, so is not informative. Conditioning on the value of $f(x)$ solves this problem.
>
> (3) Replacing $\tau$ with $f(x)$\
> The standard framework for split conformal prediction takes as input a non-conformity score $s(x,y)$ and then uses the holdout set to choose a single threshold $\tau$, which produces prediction sets of the form $P(x) =\\{y : s(x,y) \le \tau\\}$. Conformalized quantile regression by Romano et al. can also be cast in this basic framework that uses a _single_ threshold $\tau$ to create prediction sets. See e.g. the treatment of CQR in  “A Gentle Introduction to Conformal Prediction and Distribution-Free Uncertainty Quantification” by Angelopoulos and Bates: in particular equation (4) on page 8. We emphasize our framework works together with any non-conformity score, including the one used in conformalized quantile regression.
>
> (4) Lipschitzness
>
> It is true that Lipschitzness requires bounded densities, but we note that this is a mild assumption that is easy to enforce. For example, we can enforce it by perturbing the non-conformity scores with noise from a continuous distribution (e.g. uniform over $[-\epsilon,\epsilon]$) which is already standard practice (e.g. it is an option in the standard conformalized quantile regression package).
>
>
> Thank you again for engaging with interesting questions; please let us know if we can clarify anything else!

---

> ### Author Response · Authors · 2022-11-20
> **A gentle ping**
>
> Hi!
>
> We wanted to send a gentle reminder about our paper now that the discussion period has come to a close. (We know from personal experience how this kind of thing can fall by the wayside in busy periods!) If our response has addressed your questions, we would appreciate it if you could raise your score as you kindly offered to do. If not, we are very happy to answer additional questions!

---

> ### Comment · Reviewer_FUsZ · 2022-11-20
> **Response to authors**
>
> Thanks for clarifying my confusion. I think it is clear to me now after I also read the sequential version. The difference and contribution is significant where the sequential version formed as two player game and produce randomized predictor. I think this paper is worth publishing and raised my score correspondingly.

---

> > ### Author Response · Authors · 2022-11-20
> > **Thanks!**
> >
> > Thank you!

---

### Decision · Program_Chairs · 2023-01-20

**Decision:**

Accept: poster

**Justification For Why Not Higher Score:**

The paper is very hard to read. Expressions are often repeated several times, making it very messy, and very difficult to understand.

**Justification For Why Not Lower Score:**

The idea of learning a threshold function is innovative and can be applied in other contexts. The numerical demonstration is convincing and the subject in general is relevant for the community.

**Metareview: Summary, Strengths And Weaknesses:**

This work extends the scope of conformal prediction from marginal validity to conditional validity with respect to a subset of the feature space.  Two algorithms to achieve multi-valid coverage (a stronger notion than regular marginal coverage) are provided.
The main novelty lies in the fact that instead of thresholding the score functions by a constant to satisfy the prescribed coverage, the authors propose to learn a threshold function under coverage validity constraints.

**Note From Pc:**

if the above contains the word "oral" or "spotlight" please see: "oral" presentation means -> notable-top-5% and "spotlight" means -> notable-top-25%. As stated in our emails, we are disassociating presentation type from AC recommendations